# Compensating affected parties necessary for rapid coal phase-out but expensive if extended to major emitters

Lola Nacke [1], Vadim Vinichenko [1], Aleh Cherp [2,3], Avi Jakhmola [1] & Jessica Jewell [1,4,5] ✉

Coal power phase-out is critical for climate mitigation, yet it harms workers, companies, and coal-dependent regions. We find that more than half of countries that pledge coal phase-out have "just transition" policies which compensate these actors. Compensation is larger in countries with more ambitious coal phase-out pledges and most commonly directed to national and regional governments or companies, with a small share going directly to workers. Globally, compensation amounts to over $200 billion (uncertainty 163-258), about half of which is funded through international schemes, mostly through Just Energy Transition Partnerships and the European Union Just Transition Fund. If similar transfers are extended to China and India to phase out coal in line with the Paris temperature targets, compensation flows could become larger than current international climate financing. Our findings highlight that the socio-political acceptance of coal phase-out has a tangible economic component which should be factored into assessing the feasibility of achieving climate targets.

Phasing-out coal is one of the most urgent climate mitigation measures[1] and recent declines in the cost of solar and wind power[2] make it techno-economically feasible[3]. But coal phase-out risks stranding assets[4], triggering backlash from coal workers and companies[5,6], as well as causing socio-economic hardship for coal-dependent regions[7,8] and electoral losses for politicians[9]. Such challenges raise concerns about the socio-political feasibility of the rapid coal phase-out needed to meet global climate targets[10–13].

Governments are tackling these concerns with just transition strategies[7,14] which often compensate actors who bear the costs of the transition[8,15]. The idea is to increase the socio-political feasibility of coal phase-out by providing financial support to those negatively affected by it[15–17]. For example, Germany famously pledged over €40 billion to its coal dependent regions, companies, and workers as part of its coal phase-out[18,19] and coalitions of Global North countries have recently signed Just Energy Transition Partnerships (JETPs) with several emerging economies to support their coal phase-out efforts[20,21] (Supplementary Note 1). Despite their growing prevalence, there has been little quantitative or comparative analysis of such policies. Is compensation necessary for coal phase-out? How much does such compensation cost? And what type of support do compensation policies offer?

Here, we tackle these questions by systematically analyzing domestic and international coal phase-out compensation policies across four continents. These compensation policies offer a unique empirical window into the cost of making rapid coal phase-out not only techno-economically but also socio-politically feasible. By mapping coal phase-out compensation in the real-world, we are able to estimate the cost of socio-politically feasible coal phase-out.

Our analysis also sheds light on the potential cost of extending compensation to major coal consumers to accelerate coal phase-out.

[1]Department of Space, Earth and Environment, Chalmers University, Gothenburg, Sweden. [2]Department of Environmental Science and Policy, Central European University, Vienna, Austria. [3]International Institute for Industrial Environmental Economics, Lund University, Lund, Sweden. [4]Centre for Climate and Energy Transformations and Geography Department, University of Bergen, Bergen, Norway. [5]Advancing Systems Analysis, International Institute for Applied Systems Analysis, Laxenburg, Austria. ✉e-mail: jewell@chalmers.se

The Glasgow Climate Pact from COP26 (the 26th United Nations Climate Change Conference) calls for "targeted support [for coal phase-down] to the poorest and most vulnerable in line with national circumstances and recognizing the need for support towards a just transition"[22]. What financial flows would be required to expand compensation to major coal consumers for coal phase-out consistent with the Paris temperature targets? We explore this question by examining the cost of compensation to China and India, to pursue a 1.5 °C– or 2 °C– compatible phase-out if they implement compensation policies similar to those already in place in other countries. Even though neither China nor India has pledged to phase-out coal, they have the largest coal fleets which make their coal policies critical to achieving the Paris temperature targets[23].

We find that all countries which have pledged coal phase-out and have large coal power fleets also have compensation policies. We also find that the amount of compensation is generally proportional to the ambition of coal phase-out policies, which we measure as $CO_2$ emissions avoided as a result of coal phase-out pledges. Compensation policies most commonly support national and regional governments and to a lesser extent – companies involved in the transition (both coal and renewables industries), with only a small portion of funds provided directly to workers. Extending similar compensation policies for coal phase-out in-line with the Paris temperature targets in China and India would require funding equivalent to all global Official Development Assistance (ODA)[24], or roughly twice as large as international climate finance pledged under the Paris Climate Agreement[25].

## Results

### All countries with ambitious policy-driven coal phase-out have compensation policies

We construct a database of all national coal phase-out pledges and publicly-financed compensation policies (Table 1, Supplementary Table 1, Methods). We define the latter as financial transfers from governments to actors affected by coal phase-out including workers in coal power plants and mines; companies which own and operate such plants and mines; and coal-dependent countries and subnational regions (Supplementary Table 2).

We identify 43 countries which have coal phase-out pledges with specified phase-out dates and 24 countries with compensation policies (Table 1, Supplementary Fig. 1, Supplementary Table 3, Methods). With the exception of South Africa, all countries with compensation policies also plan to phase-out coal by a certain time. This means that roughly one-third of countries with coal power, covering about 16% of the global coal fleet, have both coal phase-out pledges and related compensation policies (Methods). Of the 24 countries with compensation policies, we are able to estimate the amount of compensation in 21 countries (Fig. 1, Table 1, Supplementary Table 1, Methods). For 13 cases, there is uncertainty in the amount of compensation, either because we could not confirm it in official government sources, or because the level of compensation is contingent upon future developments (such as the approval of territorial just transition plans submitted to the European Union (EU)). In such cases we provide a lower and upper estimate and use the average of the two as our central estimate (Methods). We find that globally, governments plan $209 billion in compensation for coal phase-out (central estimate; uncertainty range $163-258 billion).

We also investigate whether a more ambitious coal phase-out is associated with higher compensation. We consider a coal phase-out pledge as more ambitious when the phased-out coal power capacity is larger and/or younger and/or scheduled to be shut down faster. We operationalize ambition as avoided emissions which we calculate as the difference in cumulative emissions between a reference scenario where coal power plants are retired when they reach the average national retirement age and a scenario where coal power plants are retired in line with national pledges (Methods, refs. 23,26). Thus, the

ambition of a coal phase-out pledge reflects how large and young the prematurely retired coal phase-out capacity is as well as how fast the coal phase-out is planned to happen.

We find that all countries with large coal fleets ($\geq 20$ Gigawatt installed capacity – GW) and relatively ambitious coal phase-out pledges ($\geq 200$ Megatonne avoided $CO_2$ – $MtCO_2$) have compensation policies (Fig. 1, Supplementary Table 3). Most of these countries also have coal mining. The five countries (South Korea, Poland, Indonesia, Vietnam, and Germany) with the most ambitious coal phase-out pledges and largest coal fleets each plan compensation more than $10 billion and account for over 95% of compensation globally. Countries with smaller coal fleets ($\leq 15$ GW) and no or little coal mining plan compensation less than $2 billion (18 cases) or no compensation (20 cases – Fig. 1, Supplementary Table 3).

### Mapping funding flows and scope of coal phase-out compensation

To map the funding flows and scope of compensation, we code each individual funding mechanism within a compensation policy for the source of funds and type of support (Fig. 2, Table 1, Supplementary Table 2, Methods). About half of all compensation is international (56% central estimate; uncertainty range 43%-64%), meaning that donor countries or the EU pledge to support recipient countries in coal phase-out; the other half of compensation is domestically-funded (44%; uncertainty range 35-58%). Domestic coal phase-out compensation policies range from $0.07 billion to $66 billion (central estimates; uncertainty range $0.06-79 billion). The majority of countries with coal phase-out compensation receive some international funding though its proportion in total national compensation varies significantly (Fig. 1). Only five countries – Canada, South Korea and three EU countries without coal mining (Netherlands, Finland, and France) – do not receive any international funding. Nevertheless, the majority of international funding is received by non-EU countries (Supplementary Table 4).

We identify three international programs with provisions for coal phase-out compensation (Table 2). For recipient countries in the EU, the bulk of international funding comes from the EU Just Transition Fund (JTF) with a smaller proportion coming from the Recovery and Resilience Facility (RRF) designed to facilitate recovery from the COVID-19 pandemic. Three emerging economies (Indonesia, Vietnam, and South Africa) have signed JETPs, under which they receive funding for coal phase-out from different coalitions of Global North countries (Supplementary Note 1). Senegal has also signed a JETP agreement, but not in relation to coal phase-out, and there have been talks of a JETP with India[27], though recent reports have cast doubt on whether it will be realized[28]. We find that the annual compensation across countries generally varies between 0.001%-0.6% of GDP (Gross Domestic Product) and that domestically-funded compensation never exceeds 0.1% of GDP (Fig. 1). This suggests that there may be a ceiling within national budgets to support coal phase-out, which can be overcome through international funding.

The budget sources for both international and domestic compensation vary. The EU's JTF is funded from the EU budget 2021−2027 and 'Next Generation EU' (NGEU), its Covid recovery instrument, which also funds the RRF[29]. Domestic and sub-national funds are mobilized from a variety of sources, including regional infrastructure funds (Canada and Germany), climate and energy funds (Finland, France, and Spain), and carbon tax revenues (Canada and Greece). However, the source of funding is not always explicit. There are no official documents that specify the sources of funding in donor countries for the JETPs.

Compensation policies encompass support for five types of measures: regional development to regional authorities or SMEs (small and midsize enterprises); coal power plant and mining closure; renewables and low-carbon infrastructure development; and

**Table 1 | Coal phase-out pledges and compensation policies**

| Country | Phase-out year (previous pledge) | Compensation $billion (uncertainty) | Funding or budgetary source | Support for... |
|---|---|---|---|---|
| Germany | 2030 (2035–2038) | 66 (66–67) | Regional infrastructure fund [D] Just Transition Fund [I] | - coal power plant & mine closure - unemployment support - regional development |
| Indonesia | 2040s | 55 (31–79) 9 committed | Just Energy Transition Partnership [I] | - national government for: power plant closure, renewables & low-carbon infrastructure, unemployment support, & regional development |
| Vietnam | 2040s | 43 (25–63) 7 committed | Just Energy Transition Partnership [I] | - national government for: power plant closure, renewables & low-carbon infrastructure; unemployment support, and regional development |
| Poland | 2049 | 15 | Carbon & electricity revenues, Development fund [D] Just Transition Fund [I] | - coal power plant & mine closure - regional development |
| South Korea | 2050 | 12 (11–13) | Mobilized from treasury [D] | - renewables & low-carbon infrastructure |
| South Africa* | - | 9 | Just Energy Transition Partnership [I] | - national government for: power plant closure and regional development |
| Spain | 2030 | 2.1 | Funds managed by energy and biodiversity institutes [D] Just Transition Fund, Recovery and Resilience Facility [I] | - renewables & low-carbon infrastructure - unemployment support - regional development |
| Greece | 2028 (2023) | 1.8 (1.2–2.3) | Carbon pricing revenues [D] Just Transition Fund [I] | - regional development |
| Czechia | 2033 | 1.7 (1.2–2.2) | Just Transition Fund, Recovery and Resilience Facility [I] | - renewables & low-carbon infrastructure - regional development |
| Romania | 2032 | 1.6 | Just Transition Fund, Recovery and Resilience Facility [I] | - renewables & low-carbon infrastructure - regional development |
| Canada | 2030 | 1.2 | Green Infrastructure funding [D] Reinvesting carbon pricing revenues (regional - Alberta) | - renewables & low-carbon infrastructure - regional development - coal power plant closure - unemployment support |
| Italy | 2025 | 1.1 (0.9–1.3) | Just Transition Fund [I] | - regional development |
| Bulgaria | 2038 | 1 (0.3–1.7) | Just Transition Fund, Recovery and Resilience Facility [I] | - renewables & low-carbon infrastructure - regional development |
| Slovakia | 2025 | 0.6 (0.5–0.8) | NA [D] Just Transition Fund [I] | - coal mine closure - regional development |
| Portugal | 2021 (2030) | 0.3 | NA [D] Just Transition Fund [I] | - regional development |
| Hungary | 2029 | 0.3 | NA [D] Just Transition Fund [I] | - regional development |
| Finland | mid-2029 (2030) | 0.3 | Redirected from tendering scheme for renewables [D] | - coal power plant closure - renewables & low-carbon infrastructure |
| Slovenia | 2033 | 0.2 (0.2–0.3) | NA [D] Just Transition Fund [I] | - regional development |
| Netherlands | 2029 (2030) | 0.2 (0.1–0.3) | Ministry of Economic Affairs and Climate Policy [D] | - coal power plant closure |
| Croatia | 2033 | 0.2 (0.1–0.2) | NA [D] Just Transition Fund [I] | - regional development |
| France | 2022 | 0.1 | Program 174: Energy, climate and post-mining [D] | - regional development - unemployment support |
| Chile | 2040 | NA | NA | - coal power plant closure - unemployment support |
| North Macedonia | 2030 | NA | NA | - renewables & low-carbon infrastructure - unemployment support - regional development |
| Ukraine** | 2040 | NA | State budget | - coal power plant & mine closure |
| Total | | 209 (163–258) | Domestic: 92 (90–95) International: 117 (73–163) | - |

National time-bound coal phase-out pledges[23] in all countries with operating coal power plants at the time of making the pledge; previously pledged phase-out dates in parentheses where applicable. Central compensation estimate in $billion, uncertainty range in parentheses (Supplementary Table 3, Methods). For funding or budgetary source, domestic funds marked with [D], international funds - through the European Union mechanisms or Just Energy Transition Partnerships (JETP) - with [I]. NA indicates that the compensation amount and/or source could not be identified. Under Support for..., the five types of support in compensation policies are summarized (see also Supplementary Tables 1 and 2). *South Africa's JETP is included without an uncertainty range since there is no national coal phase-out pledge. **Ukraine's coal phase-out pledge and restructuring plan pre-dated the Russo-Ukrainian war and was thus excluded from our analysis due to the uncertainties in implementation.

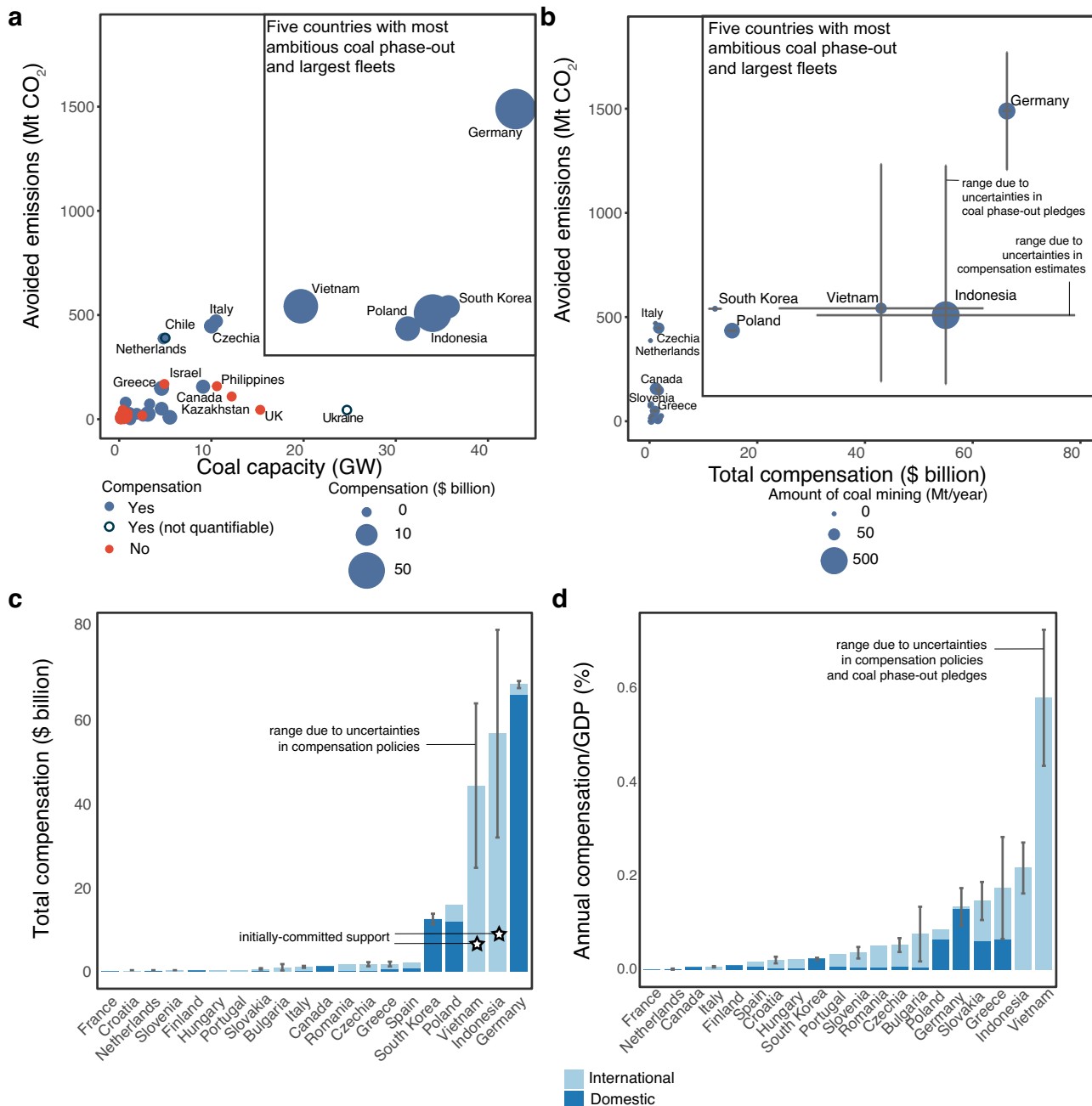

**Fig. 1 | Compensation policies in countries with coal phase-out pledges.** The central compensation estimate is represented by circle radius (panel **a**), circle position (panel **b**) or bars (panels **c** and **d**). Panel **a** shows countries with time-bound coal phase-out pledges by avoided emissions and installed capacity (Gigawatt - GW) of coal power plants[110] – blue countries with compensation policies (Supplementary Table 3) and red countries without compensation policies. Uncertainty ranges show the upper and lower estimates for compensation and avoided emissions due to uncertainties in compensation policies and coal phase-out pledges (Methods, Table 1, Supplementary Table 1). For Vietnam and Indonesia, the upper and lower estimates are based on an interpretation of the Just Energy Transition Partnerships (JETPs)[20,21] and their coal phase-out pledges[87] (Tables 1–2, Supplementary Table 3 – initially-committed support is shown is shown with a star in panel (**c**)). Panels (**b**–**d**) show all countries with time-bound coal phase-out pledges and quantifiable compensation. Panel (**b**), countries by avoided emissions from coal phase-out pledges (Megatonne (Mt) CO$_2$), total compensation, and coal mining (radius). Panel (**c**) total compensation in $ billion and (**d**) annual compensation as a proportion of national GDP (Gross Domestic Product). In panels (**c**) and (**d**), dark blue shows compensation from domestic funds, light blue from international funds. Annual compensation in panel (**d**) is calculated as total compensation divided by the number of years between the year in which the phase-out pledge was made and the pledged phase-out date.

unemployment support (Table 1, Fig. 2, Supplementary Tables 1 and 2). At the same time, the degree of specificity within compensation policies varies between countries (Methods, Supplementary Table 1). For example, while the type of support in JETP agreements is similar to other compensation policies, the JETPs do not specify the distribution between these different measures (Table 1, Supplementary Table 1).

Aside from JETP funding, over half of compensation is earmarked for regional development in coal dependent regions (central estimate $66 billion or 65% of all non-JETP funding– Fig. 2, Supplementary Table 2). The second largest amount goes to coal mining and power plant companies, across six countries (central estimate $23 billion or 23% of all non-JETP funding). Another $14 billion (central estimate, or

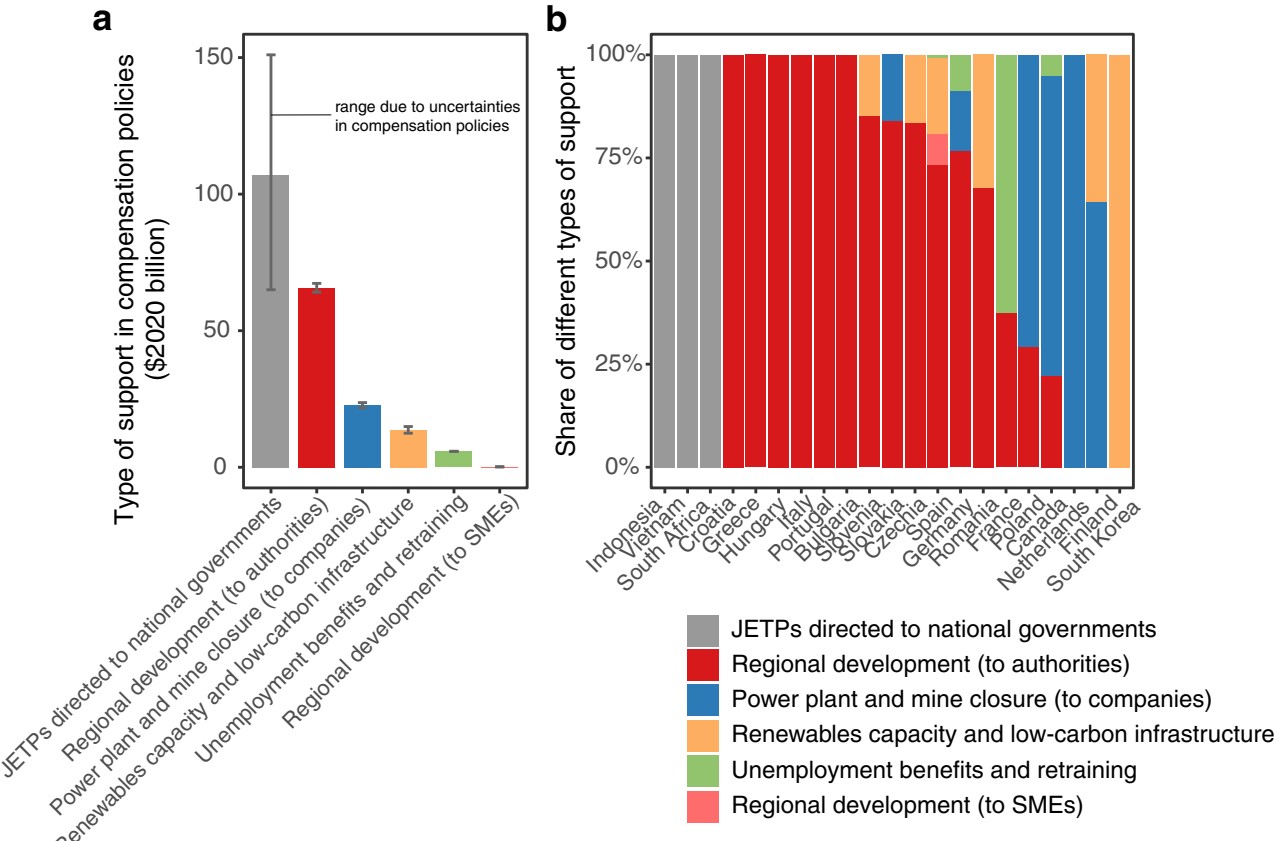

**Fig. 2 | Type of support in compensation policies.** Panel **a** Total amount of compensation by type of support in all countries with quantifiable coal phase-out compensation (Table 1, Supplementary Table 2). Bars represent central estimate and uncertainty ranges the lower and upper compensation estimates for each type of support due to uncertainties in compensation policies (Methods). All Just Energy Transition Partnership (JETP) funding is represented as funding to national governments; even though the JETP agreements list the types of support (and thus are coded in Table 1 and Supplementary Table 1), specific amounts allocated to each type of support have not been decided. Panel **b** Share for each type of support by country. Compensation under the European Union's Just Transition Fund is included under "Regional development (to authorities)", however regional actors may use it in a number of ways such as for worker retraining, coal mining closure, support for SMEs (small and midsize enterprises) and/or low-carbon infrastructure expansion. See Table 1 and Supplementary Table 1 for compensation by type of support and country and Supplementary Fig. 2 for domestic versus international funding by type of support.

14% of all non-JETP funding), goes to renewable capacity and low-carbon infrastructure expansion channeled through renewables companies and regional authorities (most notably in South Korea where all funding is earmarked for this purpose). Only four countries include direct support for unemployment benefits and worker retraining and overall this represents one of the smallest amounts of compensation (central estimate $6 billion, or 6% of all non-JETP funding). Finally, a small portion of regional compensation in one country is paid to local SMEs to support the diversification of the regional economy ($0.2 billion in Spain). For three countries (Ukraine, Chile, and North Macedonia), we were able to identify the types of support associated with compensation, but could not quantify their distribution (Table 1).

**Compensation is proportional to avoided emissions and comparable to recent EU carbon prices**

The total amount of compensation tends to be higher in countries with more ambitious coal phase-out pledges as measured by avoided emissions (Fig. 3). This is natural since early retirement of larger and younger coal power plants has wider and stronger effects on diverse interests that need to be compensated. We find that compensation is on average $37.5/ton avoided $CO_2$ emissions (range $29-$46/t$CO_2$ when accounting for uncertainties concerning coal phase-out pledges and compensation policies – Method, Supplementary Note 2).

In addition to avoided emissions, the amount of compensation is likely to be influenced by other factors which affect coal phase-out

such as vested interests[30,31], institutional capacity[26,30,32], and national wealth[23]. We thus test the relationship between avoided emissions and compensation using a multiple-variable regression analysis. We use the amount of compensation as the dependent variable (setting it equal to zero for countries with coal phase-out pledges, but no compensation policies), avoided emissions as the main independent variable and as control variables, we use variables reflecting the size and regional concentration of the coal sector, the national economic and state capacity, all of which have been shown to affect coal phase-out[23,26,30-32], as well as access to international funding (Methods, Supplementary Table 5, Supplementary Note 3). We use a series of machine-generated regression models to test the effect of different combinations of control variables as well as uncertainties arising from the ambition of coal phase-out pledges and compensation policies (Methods, Supplementary Tables 12–16, Supplementary Notes 2, 3).

We find that avoided emissions is the most consistent and strongest predictor of compensation – present at a significance level of $p < 0.1\%$ in our 50 best-performing models (Supplementary Tables 12–16). The co-efficient for avoided emissions within our best-performing models ranges from $27-45 per ton of avoided $CO_2$ emissions which is similar to the directly calculated compensation per ton of avoided $CO_2$ emissions ($29-46/t$CO_2$) (Supplementary Note 2, Supplementary Table 9). This shows that compensation amounts are generally proportional to the ambition of coal phase-out pledges also when controlling for the strength of the coal sector, state capacity, and

**Table 2 | International coal phase-out compensation mechanisms**

| Compensation program | Total amount $billion (uncertainty) | Recipients | Donors | Description |
|---|---|---|---|---|
| Just Transition Fund (EU JTF) | 9 (7–11) | Bulgaria, Croatia, Czechia, Germany, Greece, Hungary, Italy, Poland, Portugal, Romania, Slovakia, Slovenia, Spain[68,69] | EU | EU support for Member States with regions vulnerable to negative effects of the transition to a climate-neutral and circular economy. |
| Recovery and Resilience Facility (EU RRF) | 1 | Bulgaria, Czechia, Spain, Romania[76] | EU | EU support for Member States to recover from the impact of the Covid pandemic and make their economies more resilient and sustainable. |
| Just Energy Transition Partnership (JETP) | 55 (31–79) 9 committed | Indonesia[21] | Canada, Denmark, EU, France, Germany, Italy, Japan, Norway, UK, US | Support the achievement of Net Zero by 2050 and transitioning away from on- and off-grid coal-powered electricity. Currently committed over the next 3-5 years, with potential for "policy reforms aimed at facilitating greater levels of investment"[21]. |
| | 43 (25–63) 7 committed | Vietnam[20] | Canada, Denmark, EU, France, Germany, Italy, Japan, Norway, UK, US | Support the achievement of Net Zero by 2050, and the transition away from fossil fuels. Currently pledged over the next 3-5 years. "The continuation of the partnership is expected to be contingent on [conditions outlined in the JETP]"[20] |
| | 9 committed | South Africa[64,67] | EU, France, Germany, UK, US | Accelerate decarbonization, with a focus on the electricity system, and help achieve South Africa's NDC. Currently committed "over the next 3-5 years [...,] with a view to longer term engagement"[64]. |
| All JETP funding | 107 (65–151) 25 committed | | | |
| All international funding (JETP + EU) | 117 (73–163) | | | |

All international programs funding coal phase-out compensation. Central estimates for all programs are shown in bold. For the European Union's Just Transition Fund (JTF) and Recovery and Resilience Facility (RRF), only funding likely to support coal phase-out in recipient countries is included (Methods). The Just Energy Transition Partnerships (JETPs) aim to support decarbonization in the recipient countries, with phasing down coal power-generation as one of the main objectives. The central estimate and uncertainty range for the JETPs are based on an interpretation of the Partnership Agreements and coal phase-out pledges associated with these countries (Methods, Supplementary Notes 1 and 2). Such a calculation is not possible for South Africa because the country has not pledged to phase-out coal by a certain date.

access to international funding. While these observations cover a third of all countries which use coal power, both estimates should be treated with a degree of caution given the relatively limited number of cases in our regression analysis.

Both the direct and regression-based estimates for the average compensation per ton of avoided emissions are well within the range of the carbon price under the European Union Emission Trading Scheme (EU ETS) over the last five years (Fig. 3). Countries with compensation below the carbon price tend to have no active coal mining (e.g. Italy, France and the Netherlands) or a particularly small coal fleet (e.g. Slovenia). Hungary is a clear outlier – while its total compensation is comparable to Finland and Portugal, its coal phase-out affects one coal plant which has already been in operation for more than 50 years and a small coal mining industry, and thus the avoided emissions are very low[33,34].

We also calculate the average compensation per GW of installed coal capacity across all countries as $0.8 billion/GW (uncertainty range $0.7-1.1 billion/GW), which is generally below the cost of new coal power capacity in Europe (Fig. 3). This is to be expected since most countries plan to retire aging power plant fleets, which have already depreciated in value. The impact of the age of coal power plant fleets is likely why avoided emissions is a better predictor for compensation than installed capacity. Yet, there are three cases where compensation per GW is comparable to or exceeds the cost of a new coal plant – Indonesia, Vietnam, and Germany. In the case of Vietnam and Indonesia, this likely reflects the relatively young coal fleet, large coal power plant pipeline[35], and distant phase-out. In the case of Germany, while the overall compensation is greater than the cost of a new coal power plant, compensation to companies supporting power plant and mining closure is less [also see ref. 36]; nevertheless, some have criticized the German government for overcompensating companies relative to the value of the retired coal plants[37].

## Compensation for coal phase-out in China and India would outstrip existing climate finance

China and India are the two countries with the world's largest coal fleets and would need to phase out coal within the next twenty years for the Paris Agreement temperature targets, which is faster than other major coal users[13,23] (Table 3). However, China is still expanding its coal fleet with more than 200 GW of new coal capacity in the pipeline[38] in spite of a pledge to slow coal expansion and "start phasing down coal use from 2026"[39]. Likewise, India has not set a date for coal phase-out[40].

We project the compensation required for China and India to phase out coal in a counterfactual scenario under which both countries phase out coal in line with the Paris temperature targets, and both adopt compensation policies similar to those currently implemented in other countries. First, we calculate the avoided emissions for China and India in 1.5 °C-compatible and 2 °C-compatible pathways. For 1.5 °C-compatible pathways we use the C1 and C2 categories in the IPCC AR6 (the Sixth Assessment Report of the Intergovernmental Panel on Climate change[1]) and for 2 °C-compatible pathways the C3 and C4 categories (Methods). We find that the avoided emissions for China and India combined under 2 °C-compatible pathways are 11 times higher than avoided emissions implied by today's coal phase-out pledges worldwide, and avoided emissions under 1.5 °C-compatible pathways are 15 times higher (central estimates) (Table 3). Subsequently, we project compensation for China and India based on these avoided emissions. For the central compensation estimate, we use the median avoided $CO_2$ emissions in IPCC AR6 pathways multiplied by the average compensation per ton avoided emissions in countries with time-bound coal phase-out pledges (Fig. 3). To assess the uncertainty, we use (1) the interquartile range of avoided $CO_2$ emissions in IPCC AR6 pathways and (2) the range of compensation per ton avoided emissions from our direct-calculation as well as our 50 best-performing regression models which capture different combinations

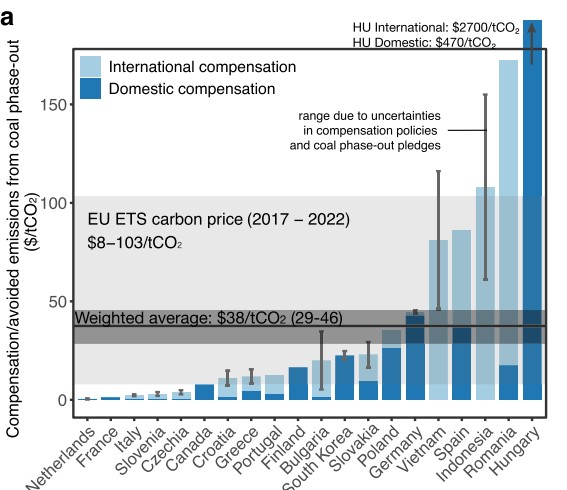

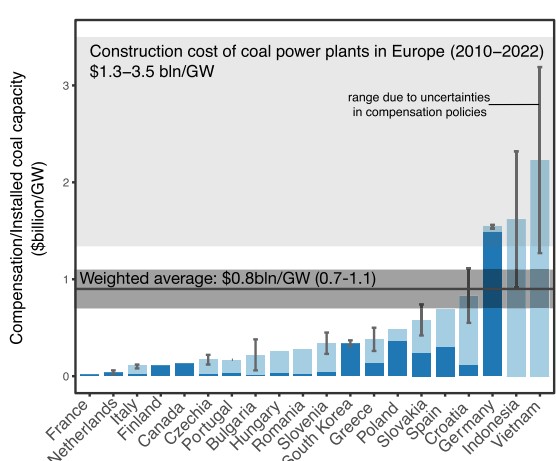

**Fig. 3 | Compensation per avoided emissions and coal power capacity.** Bars represent the central compensation estimate and uncertainty ranges the lower and upper estimates due to uncertainties in compensation policies and coal phase-out pledges (Methods, Table 1, Supplementary Tables 1 and 3). Dark blue, compensation from domestic funds; light blue, international funds. Panel (**a**) Compensation normalized to tons of avoided emissions from coal phase-out pledges (Methods). The light gray band shows the carbon price under the European Union (EU) emissions trading scheme over the past five years[106]. The black line and dark gray band show the average compensation per ton avoided $CO_2$ emissions (central estimate and uncertainty range – Supplementary Note 2). Panel (**b**) Compensation normalized to gigawatt (GW) of installed coal capacity at the time of taking the coal phase-out pledge. The light gray band shows the construction cost of coal power plants in Europe over the last ten years (see Supplementary Table 6 for construction costs by power plant). Black line and dark gray band show the average compensation by installed coal capacity (central estimate and uncertainty range – Methods, Supplementary Table 10, Supplementary Note 2).

of control variables and confidence intervals (Supplementary Tables 10, Supplementary Note 2).

We find that compensation for China would be $2.2 trillion (central estimate; uncertainty range $1.2–5.3 trillion) under a 1.5 °C-compatible pathway and $1.6 trillion ($1.1–4.8 trillion) under a 2 °C-compatible pathway; while compensation for India would be $1 trillion ($0.5–1.6 trillion) under a 1.5 °C-compatible pathway and $0.8 trillion ($0.5–1.3 trillion) under a 2 °C-compatible pathway – Table 3. China has experience with a similar policy from a support scheme for the re-employment of workers in heavy industries who lost jobs due to overcapacity[41], however, the overall cost of this measure was roughly two orders of magnitude less than the potential amount of compensation we calculate as needed for 2 °C-compatible pathways in China. Recycling funds currently allocated to coal production subsidies in China[41] and India could theoretically offer a funding source for such compensation, however in both countries these subsidies are still far smaller than the estimated compensation needed for coal phase-out compatible with the Paris temperature targets (Fig. 4, Supplementary Fig. 3).

Given the high cost for China and India, international funding might be required for coal phase-out compensation (as ref. 42 argues for India) though there is no such negotiation process for China and recent reports indicate that a JETP-type agreement between India and donor countries is unlikely[28]. The two compensation policies which can be considered the closest analogues to policies for India and China are the JETP agreements with Indonesia and Vietnam – both of which are emerging economies and major coal consumers. However, funding 1.5 °C– or 2 °C– compatible coal phase-out in India would require roughly 10 times as much JETP funding as Indonesia and Vietnam combined are likely to receive and compensation for China would require roughly 20 times as much as implied by existing JETPs (assuming today's JETPs continue through the pledged coal phase-out – Methods, Supplementary Fig. 4, Supplementary Table 7). Furthermore, we find that compensation for India would be roughly 5 times larger than all existing compensation worldwide and for China roughly 11 times larger.

Funding coal phase-out compensation policies in China and India through international support would also require mobilizing significantly higher levels of international assistance. The funding for JETPs in South Africa, Indonesia and Vietnam is comparable to the Official Development Assistance (ODA)[43] these countries receive – Fig. 4. While the JETPs don't explicitly rely on ODA, such assistance is sometimes earmarked for climate purposes[24] and is indicative of existing financial transfers between countries. In contrast, in India, annual funding required for coal phase-out in line with the Paris temperature targets is at least 10 times the amount it receives in gross ODA, and in China roughly 60 times. In fact, in 2021, the total ODA globally was $176 billion[24] (Methods) – roughly comparable to the annual estimated coal phase-out compensation for China and India for climate targets. Coal phase-out in these two countries alone would require almost all of the COP climate finance pledged by developed countries[25] (Fig. 4, Supplementary Fig. 4) though climate finance is meant to support not only climate change mitigation but also adaptation efforts[25] and not only India and China.

We also tested how our conclusions would change under a less ambitious climate scenario, which is more consistent with the level of ambition implied by today's coal phase-out pledges (Supplementary Figs. 3–5, Supplementary Table 7). Today's coal phase-out pledges are generally considered to be too weak to be compatible with Paris temperature targets[23,44] and instead are compatible with keeping warming below 2.5 °C[23]. To reach a level of ambition similar to existing coal phase-out pledges (i.e. in line with 2.5 °C-compatible pathways – or category C5 in the IPCC AR6 pathways), avoided emissions in China and India combined would be 4 to 11 times higher than avoided emissions from all current coal phase-out pledges combined and compensation would be roughly 15 times more than compensation implied by all existing compensation policies (Table 3, Supplementary Table 7). Compared to ODA, in India, estimated annual compensation in line with 2.5°C-compatible coal phase-out is still roughly 4 times the amount it receives, and in China roughly 26 times (Supplementary Fig. 5, Supplementary Table 7). Together, estimated annual compensation for China and India under a 2.5 °C-compatible pathway would

**Table 3 | Coal phase-out pledges and compensation policies compared to coal phase-out in China and India compatible with the Paris temperature targets**

| Region/Country | Compensation estimate | | Gt CO₂ avoided emissions | | Coal phase-out pledge (year(s)) | | Net zero target(s) |
|---|---|---|---|---|---|---|---|
| Compensation in countries with coal phase-out pledges and compensation policies | | | | | | | |
| JETP countries | $0.1 trn [0.07–0.15] | | 1.1 [0.4–2.5] | | 2040 s for Indonesia and Vietnam (South Africa no pledge) | | 2050 |
| non-JETP countries | $0.11 trn [0.1–0.11] | | 4.7 [4.3–5.2] | | 2021–2050 | | 2035–2050 |
| Total | $0.2 trn [0.16–0.26] | | 5.8 [4.7–7.7] | | - | | - |
| Pathway-based estimates of compensation for different temperature targets | | | | | | | |
| | 1.5 °C | 2 °C | 1.5 °C | 2 °C | 1.5 °C | 2 °C | |
| China | $2.2 trn [1.2–5.3] | $1.6 trn [1–4.8] | 60 [42–69] | 43 [36–57] | 2035 [2030–2040] | 2040 [2035–2045] | 2060 |
| India | $1 trn [0.5–1.6] | $0.8 trn [0.5–1.3] | 26 [21–30] | 21 [17–25] | 2038 [2035–2041] | 2045 [2040–2045] | 2070 |
| Total (China and India) | $3.2 trn [1.7–6.9] | $2.4 trn [1.9–6.1] | 86 [63–99] | 64 [53–82] | - | - | - |

Central compensation estimate for countries with coal phase-out pledges and compensation with uncertainty range in brackets (Table 1, Supplementary Table 3). For Just Energy Transition Partnership (JETP) countries, compensation and net zero targets include Indonesia, Vietnam and South Africa, while avoided emissions and coal phase-out pledges only include Indonesia and Vietnam (South Africa does not have a coal phase-out pledge – Methods, Supplementary Note 1). Data for Indonesia and Vietnam based on their coal phase-out pledges under the Global Coal to Clean Power Statement[87]; compensation estimates for Indonesia and Vietnam based on an interpretation of the JETPs[20,21]. Net zero targets for JETP-countries are from the JETP agreements. Non-JETP countries include all other countries with time-bound coal phase-out pledges and quantifiable compensation policies (Table 1, Supplementary Tables 1 and 3). Net zero targets for non-JETP countries based on ref. 23. For China and India, pathway-based estimates of compensation for different temperature targets include a central estimate based on the average compensation per ton avoided emissions ($37.5/tCO2) and median avoided emissions from the Intergovernmental Panel on Climate Change's Sixth Assessment report (IPCC AR6) pathways. 1.5 °C based on IPCC AR6 C1- and C2-categories; 2°C, on C3- and C4-categories. (See Supplementary Table 8 for individual IPCC AR6 pathway categories, and Supplementary Table 7 for 2.5 °C-compatible estimates based on category C5). Uncertainty ranges for pathway-based compensation estimates are based on the range of average compensation per ton avoided emissions, the different sets of regression analyzes, and the interquartile range of avoided emissions under IPCC AR6 pathways[1] (Supplementary Tables 12–16, Methods, Supplementary Note 2). Uncertainty range of avoided emissions and coal phase-out pledge for China and India based on interquartile range of respective IPCC AR6 pathways. The date of coal phase-out for China and India is calculated in each pathway as the date when unabated coal power generation falls below 1% of the total electricity supply (Methods, ref. 23).

amount to more than half of the $100 billion annual climate finance target pledged under the Paris Agreement[25]. Thus, while compensation for China and India for 2.5°C-compatible pathways is much more affordable, it would still pose a significant financial burden on the international system.

## Discussion

Our results have direct implications for both domestic policies and international agreements on just energy transitions. Among diverse policies to facilitate coal phase-out[32,45], our research suggests that compensating affected actors is essential, especially in the case of large-scale and rapid phase-out. We also find that as a rule, the more ambitious the coal phase-out pledge, the higher the compensation is. At the same time, the level of compensation per ton of avoided emissions is comparable to recent carbon prices within the EU ETS. This casts compensation as not only necessary but also a rational policy because it suggests that societies can either pay to emit $CO_2$ or cut emissions and compensate affected actors with approximately the same price tag, the latter option being more attractive for the climate.

Compensation policies can be justified not only on the basis of political expediency, but also on ethical grounds of ensuring 'just transitions'[14,46,47]. Our finding that the amount of compensation is proportional to avoided emissions indicates that compensation policies are not spurious, but rather seek to address the negative impacts of coal phase-out, which are likely to be larger in the case of faster and wider retirement of coal power plants. Yet, the existence of compensation policies by no means ensures a just coal phase-out. We show that the main beneficiaries of compensation policies are coal-dependent countries, regions, and energy companies rather than workers. Future research should investigate to which extent the distribution towards larger actors and entities aligns with principles of distributional, restorative, procedural, and recognitional justice (Supplementary Note 4)[18,48–51] and thus more closely engage with

arguments that view compensation as merely support for fossil fuel interests[19].

Two particularly pressing policy questions, as just transition policies shift from formulation to implementation, is how to ensure good governance of such policies and to what extent compensation covers the real costs for different actors (Supplementary Note 1). These costs can be assessed by bottom-up analyzes as in ref. 42 which provides estimates for India roughly comparable with our results (Supplementary Table 9), but more research is needed to compare such bottom-up and top-down estimates.

Our findings also signal what it would take to extend compensation policies, which are largely limited to Europe and JETP countries, to the world's major coal consumers, if the latter phase out coal as fast as required by the Paris temperature targets. We show that in such a scenario, compensation in China and India would likely need to be funded in large part from international sources and could pose a significant financial burden on international climate financing. More precisely, it would likely exceed both the Paris climate finance pledge and potentially all global Official Development Assistance.

Our analysis covers all compensation policies in countries with time-bound coal phase-out pledges (which is a third of all countries with coal power). Nevertheless, as these policies diffuse to new countries, the analysis should be continuously updated as the mechanisms affecting compensation could shift. For one, the willingness of donor countries to fund coal phase-out and/or the willingness of affected actors to accept certain levels of compensation could change. Today's compensation policies may also send signals either encouraging or discouraging coal power expansion depending on whether the relevant governments and companies expect to be donors or recipients of future compensation. Future work should also examine the role of compensation in market- rather than policy-driven coal decline. Our preliminary review suggests that even in the absence of deliberate coal phase-out policies, coal decline has often been accompanied by compensation to affected communities[8,52], but

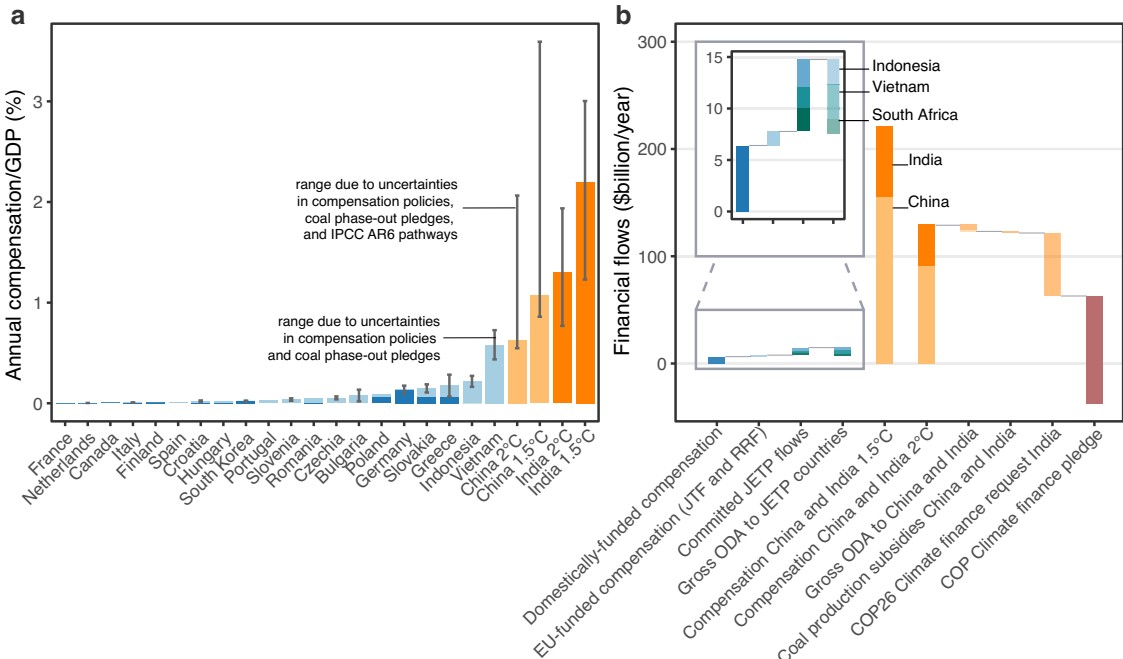

**Fig. 4 | Coal phase-out compensation for China and India for Paris temperature targets in context.** Annual compensation as a proportion of GDP (Gross Domestic Product) (panel **a**) and annual compensation in $billion compared to other international financial flows (panel **b**). For countries with compensation policies, annual compensation is calculated as the total compensation divided by the number of years between the phase-out pledge and the pledged coal phase-out (and in the case of panel a normalized to GDP[84] in the year each phase-out pledge was made). For China and India, annual compensation is calculated for each individual 1.5 °C- or 2 °C-compatible pathway and normalized to GDP in 2021[84] (Methods). 1.5 °C estimates include C1- and C2-categories from the IPCC AR6 (Intergovernmental Panel on Climate Change's Sixth Assessment Report); 2 °C include IPCC AR6 C3- and C4-categories. (See Supplementary Table 8 for compensation estimates based on individual categories and Supplementary Table 7 for 2.5 °C-compatible estimates). In panel (**a**), blue bars represent the central compensation estimate for all countries with time-bound coal phase-out pledges and quantifiable compensation – dark blue, from domestic funds; light blue, international funds – and uncertainty ranges the lower and upper estimates due to uncertainties in compensation policies and coal phase-out pledges (Methods, Table 3). Orange bars represent the central pathway-based estimates of potential coal phase-out compensation for India and China to stay on 1.5 °C– and 2 °C–compatible pathways with the uncertainty ranges representing uncertainties based on the range of average compensation per ton avoided emissions, the regression analyzes, and the interquartile range of avoided emissions under IPCC AR6 pathways. Panel (**b**) shows central estimates for domestic (dark blue bars) and international (light blue and green bars) compensation compared to gross Official Development Assistance (ODA)[43], pathway-based compensation (orange bars), coal production subsidies in China and India[108], the climate finance request by India's Prime Minister at COP26 (26th Conference of the Parties)[109] (annualized from 2023 to the year of the median coal phase-out date – Methods), and the climate finance pledge first made at COP15 and recently recommitted to at COP26[25] (Methods, Supplementary Table 7).

more work is needed to systematically identify and examine such policies.

More generally, our research serves as a model for quantifying the social and political concerns of rapid transitions[53,54] so that they can be considered on par with economic aspects, and ultimately integrated in climate-energy models[55,56]. In spite of the cost-effectiveness of coal phase-out, its feasibility is often challenged on socio-political grounds[10,13,23,26,57]. We show that political will and social acceptance have a tangible economic component that can be at least partially quantified in monetary terms. This approach can be used to better estimate the real costs of climate policies to governments and could be extended to other climate actions mired by socio-political resistance.

## Methods
### Identifying countries with coal phase-out pledges and compensation policies
We build a database of national coal phase-out pledges and compensation policies combining systematic document review, web searches and expert consultations (Supplementary Fig. 1).

First, we identify all countries with a time-bound coal phase-out pledge and installed coal capacity at the time of making the pledge (Supplementary Table 3)[23]. This database was built through a systematic review of national and international documents including National Energy and Climate Plans (NECPs); National Recovery and Resilience Plans (NRRPs); Nationally Determined Contributions

(NDCs); the Powering Past Coal Alliance (PPCA); the Global Coal to Clean Power Statement (GCCP); and other policy documents. It covers all explicit national coal phase-out pledges, but does not include coal phase-out implied by national net-zero or other climate targets since such plans may or may not feature coal phase-out. It also excludes countries which have joined the PPCA, but where there is no official date associated with the coal phase-out pledge (the United States[58], Kosovo[59], and Mexico[60]).

Second, we identify all compensation policies in these countries using a systematic Google search and the terms "coal phase-out", "coal", "just transition" and "coal compensation". We identified 23 countries that have both coal phase-out pledges and related compensation policies. To confirm our case identification, we consulted experts in two surveys and three workshops. The first survey was conducted in September 2021 with a selection of 15 coal phase-out experts. The second was conducted in January/February 2022 and distributed to the same 15 experts as well as via Twitter (now called X). We received 14 expert responses. In both versions of the survey, we presented respondents with our criteria for case selection (explicit coal phase-out pledge and a compensation policy) and our initial set of cases, and asked them two questions: (1) Are you aware of any governments not included in the list above that are planning to phase out coal and compensate affected actors? (2) Are you aware of any other governments that compensate affected actors of coal sector declines without a deliberate coal phase-out policy?

We asked the same two questions to attendees at three online workshops on fossil fuel decline – two associated with the CINTRAN project[61] and one associated with the Contractions project[62]. Through the surveys and expert consultations, we found that while Poland has not finalized its law, the country has plans for compensation[63]. The study was exempt from ethical review and approval subject to Swedish higher education regulation, because no sensitive personal data were collected or processed as part of the surveys and the workshops.

We also include compensation for coal phase-out through the EU Just Transition Fund (JTF) and the Just Energy Transition Partnerships (JETPs), both of which were announced during our analysis. At the time of writing, three countries have JETPs connected to the phase-out or phase-down of coal power: South Africa[64], Indonesia[21] and Vietnam[20] (Supplementary Note 1). Indonesia and Vietnam have coal phase-out pledges, but South Africa does not. We thus include the JETP for South Africa in our analysis (Table 1, Table 3), but cannot consider it as a case in our average compensation per avoided tCO$_2$ emissions or regression analysis.

To calculate the global share of countries with coal power and compensation policies, we used data on installed coal capacity by country from ref. 65. We consider all countries with installed coal capacity >100 MW (77 countries in 2022). The global installed capacity in 2022 was 2084 GW.

## Quantifying and mapping financial compensation for coal phase-out

We code each compensation policy for: the amount of compensation; the type of support; and the funding or budgetary source from which compensation is paid (Supplementary Table 1 and ref. 66).

We only consider public finance to enable consistent comparison across cases and since private investment is likely to follow another logic.

In the majority of cases we rely on official governmental (laws, national budgets, strategies, plans and press releases) and international sources (the JETPs[20,21,67], the EU JTF Allocation[68,69], NRRPs[70–72], and EU case law[73,74]) – Supplementary Fig. 1. We identify government sources by searching national and ministerial websites for each of our national cases with the search terms "coal phase-out", "coal", "just transition" and "compensation". We use these terms in the English version of government websites, and where English versions are not available, we translate terms into the national language, and the search results into English using Google translate.

In four cases (Poland, Netherlands, Greece, and for Germany's auction system) we found evidence of compensation in third-party sources but could not identify a government source (Supplementary Table 1), but in the case of Poland, could confirm the existence of the compensation policy through the expert consultation.

For the JETP agreements, we include South Africa, Indonesia, and Vietnam in our analysis but not Senegal since the Senegalese JETP does not refer to coal phase-out[75]. The EU JTF includes support for regions in EU member states expected to be negatively affected by climate mitigation measures including coal phase-out[69]; we use the Territorial Just Transition Plans (TJTPs) and other European Commission documentation to estimate how much of this funding is likely to be related to coal phase-out[69]. We also include NRRPs because the Recovery and Resilience Facility supports not only economic recovery from the Covid 19 pandemic, but also low-carbon transitions[76]. NRRPs explicitly describe what measures funding is intended for so we include amounts related to coal phase-out.

In 13 cases there is uncertainty about the amount of compensation either because we could not verify part of the compensation policy in government sources; in the case of the EU JTF it was not possible to identify how much funding relates to coal phase-out versus other climate change mitigation measures; and in the case of Vietnam and Indonesia the JETPs are only pledged for an initial period.

We capture these uncertainties with a lower and upper estimate for compensation (Table 1, Supplementary Table 2). The lower estimate includes: only funding mechanisms we could confirm in official government sources; for the EU JTF, an estimate of the likely share of overall funding based on nationally-specific documentation (Supplementary Table 3); and for JETP countries, a plausible lower estimate based on the JETP agreements and an earlier coal phase-out (Supplementary Notes 1 and 2). The upper estimate includes: funding mechanisms we could only confirm in third-party sources; for the EU JTF, the entire amount of compensation pledged or in the case of Bulgaria all JTF funding since at the time of writing, the country's TJTP had not been approved (Supplementary Table 1); and for JETP countries, a plausible upper estimate based on the JETP agreements and a later coal phase-out (Supplementary Notes 1 and 2).We calculate a central estimate as the mean of the upper and lower estimates.

We excluded three specific funding mechanisms for which the situation has substantially changed since the compensation policy was announced. For Germany we excluded potential compensation to electricity consumers dependent on future electricity price changes due to the coal phase-out since the policy was formulated prior to the Russo-Ukrainian war and the resulting energy crisis; given the current situation it is difficult to estimate compensation in this context and highly uncertain whether it should be attributed to coal phase-out. In the Netherlands we excluded requests for compensation from two coal power plant owners since they have been struck down by the courts[77]. We also excluded Ukraine from our analysis. While Ukraine declared coal phase-out in 2020 and specified compensation to coal companies in its 2022 budget[78], the start of the war in February 2022 has made implementation of these plans highly uncertain.

We could not quantify compensation for two countries: Chile pledges to compensate power companies based on a capacity mechanism but does not specify how this capacity mechanism will be calculated[79–81]. North Macedonia's NECP mentions funds for coal phase-out and a just transition but has not yet specified the amount or funding source[82].

All compensation is reported in US Dollars (USD) 2020 using the exchange rate[83] and Gross Domestic Product (GDP) deflator[84] for the respective year and country. We used the year in which the compensation is numerated in, if specified, or the year in which the respective document was published.

We calculated the average for compensation per ton of avoided emissions using the total compensation divided by the sum of avoided emissions in all countries with compensation policies total. We replicated this calculation using both an optimistic and pessimistic interpretation of the coal phase-out pledges and the upper and lower estimates for compensation (Supplementary Table 10).

We identify the funding source and type of support implied by compensation policies based on the national and international documents described above and government budgets. We retrieve government budgets for the years of and following coal phase-out pledges (for example, France's coal phase-out pledge was made in 2017 so we use government budgets from 2017-2021). We search government budgets for the term "coal" (or its equivalent in the national language, for example "charbon" in French) and code budget entries specifically related to coal phase-out. To ensure transparency over how we code compensation by the scope or type of support, we provide a "description of support" for each individual funding mechanism in Supplementary Table 1.

## Calculating avoided emissions from coal phase-out

We follow the method developed in refs. 23,26 to calculate avoided emissions resulting from national coal phase-out pledges[85], and for India and China under 1.5 °C-, 2 °C- and 2.5 °C-compatible pathways[1,86] from the Sixth Assessment Report of the Intergovernmental Panel on Climate Change (IPCC AR6), versus a reference scenario where coal plants are retired following the average historical national lifetimes.

We calculate the average historical lifetime of coal power plants since 2001 in each country using the S&P database[65]. If a country has fewer than four retirement events in that period, we use the global average lifetime (42 years), except for Asian countries where we use a regional average which is markedly shorter (30 years)[23,26].

For the reference scenario, coal plants are retired using truncated normal distribution based on the historical national lifetime and its standard deviation (see ref. 23,26). For this calculation, we assume that no new plants are built beyond those already under 'construction' at the time each phase-out pledge was made. We assume that plants currently under construction come online at the planned date specified for each plant in ref. 65. In the reference scenario, coal plant retirements begin in 2022 for countries who recently adopted coal phase-out pledges or in 2018 for countries which adopted coal phase-out pledges prior to 2019 (see ref. 26). This is done to best capture the expected effect of the pledges at the moment of taking them.

In the phase-out scenario, we assume that plants follow a natural retirement trajectory – that is they follow the same logic as the reference scenario – until the pledged retirement date, when all remaining plants are abruptly retired. For countries with ranges in their coal phase-out pledge, we calculate several phase-out scenarios. For example, Vietnam and Indonesia pledge to phase out coal "in the 2040s"[87], thus we assume an optimistic phase-out by 2040, a central phase-out by 2045, and a pessimistic phase-out by 2049. For Germany, the phase-out scenarios correspond to the multi-stage coal phase-out plan proposed by the Coal Commission, and the new phase-out year envisioned by the current government[88] (Supplementary Text 4 in ref. 26).

We calculate the avoided emissions from coal phase-out as the difference in emissions between the reference and the coal phase-out scenarios for each country. We do this by multiplying the capacity of prematurely retired coal power plants by the number of years between the retirement under the reference and coal phase-out scenarios and accounting for the historical national load factor as well as technology-specific efficiencies and emission rates for the thermal content of different coal types to convert avoided generation into avoided emissions (see ref. 26 for more details).

For avoided emissions of China and India under 1.5 °C-, 2 °C- and 2.5 °C- compatible IPCC AR6 pathways we use a similar methodology and calculate the difference between coal emissions under a reference scenario and estimated coal emissions in IPCC AR6 pathways from 2022, essentially seeing the former as a carbon budget for coal generation (see also ref. 23). For the reference scenario, we calculate emissions from coal power for all countries in the China+ and India+ regions from the set of ten regions (R10) using the same approach we describe above.

We then calculate unabated coal power generation under 1.5 °C-, 2 °C- and 2.5 °C- compatible pathways as the difference between total electricity generation from coal (variable "Secondary Energy|Electricity|Coal") and generation from coal with CCS ("Secondary Energy|Electricity|Coal|w/ CCS").

For 1.5 °C-compatible pathways, we use categories C1 (no/low overshoot) and C2 (high overshoot), for 2 °C-compatible pathways categories C3 (likely below 2 °C) and C4 (below 2 °C) and for 2.5 °C-compatible pathways we use category C5 (below 2.5 °C). We interpret AR6-scenarios in line with ref. 23 which includes pathways which return to 1.5 °C after a high overshoot as 1.5 °C-consistent (corresponding to C2 pathways in AR6) and 'higher-2C' pathways as 2 °C-consistent (corresponding to C4 pathways). This approach follows the original formulation of 1.5 °C -consistent pathways as those with and without overshoot[89] or both C1 and C2 category pathways[1]. This approach is on the broader end of a range of different interpretations in the literature. On the narrower end of interpretations, some view only a subset of C1-pathways as Paris-consistent[90] and others a subset of C2-C3-pathways[91] as 1.5 °C-consistent. Since we take a broader approach, we use the term 1.5°C-compatible rather than 1.5 °C-consistent. We test the effect of using individual pathways for 1.5 °C- and

2 °C-compatible compensation and find that this does not significantly affect our results (Supplementary Table 8).

We convert unabated coal generation to emissions using the same emission intensity as in the reference scenario for the respective region in order to estimate required avoided emissions under each 1.5 °C-, 2 °C- and 2.5 °C- compatible IPCC AR6 pathway.

We also estimate the coal phase-out year in the China+ and India+ regions in climate mitigation pathways with the median (and interquartile range (IQR)) of the first reported year when unabated coal power generation falls below 1% for each region across the sets of 1.5 °C-, 2 °C- and 2.5 °C- compatible pathways respectively (see Table 3).

## Multivariable regression analysis

We conduct a multivariable regression analysis to measure the relationship between coal phase-out ambition and compensation while controlling for variables reflecting characteristics of the coal sector and the national context which are likely to affect compensation. We identify these variables based on theoretical and empirical evidence from previous literature (Supplementary Note 3).

Our sample includes all countries with coal phase-out pledges and installed coal capacity at the time of making the pledge (Supplementary Table 3), and for which data on compensation and all control variables were available. We could not quantify compensation for Chile, Ukraine or North Macedonia and there was no state capacity data for Brunei Darussalam which resulted in a sample of 39 countries.

Our outcome variable is the central estimate of coal phase-out compensation. For countries with a coal phase-out pledge but no compensation (20 countries, Supplementary Table 3), we set compensation equal to zero.

We group our independent variables in six categories representing similar mechanisms:

First, variables related to the ambition of national coal phase-out. This includes (1) Avoided emissions (Megatonne (Mt) $CO_2$). We use three sets of avoided emissions estimates – a central estimate, an optimistic estimate where coal is phased-out at the earliest possible date, and a pessimistic estimate where coal is phased-out at the latest possible date (Supplementary Table 3) based on our own calculation as described above. (2) Number of years over which coal phase-out is pledged based on our own calculation as the difference between the year in which each phase-out pledge was made, and the end-year by which coal is pledged to be phased-out.

Second, control variables related to the strength of the coal sector or third, the level of vested interests. This includes (1) Installed capacity of operating coal power (Gigawatt (GW)) in the year of the phase-out pledge based on ref. 65. (2) Average coal power generation (2016-2020) based on ref. 92. We use an average since coal power generation fluctuates due to e.g. energy demand changes or availability of other electricity generation sources (3) Average coal mined (Mt) based on ref. 93 for most countries, ref. 94 for Greece, Bulgaria and Slovakia, and ref. 95 for Vietnam. We use the average over the last five available years due to fluctuations in annual coal production. (4) Coal share in power generation based on data from ref. 92, for the year in which coal phase-out was pledged. (5) Number of coal workers; own calculation based on national employment factors from ref. 96, multiplied by installed coal capacity, coal capacity in construction, and amount of coal mining (see sources above), respectively. (6) A variable measuring the regional concentration of coal power plants within a country. Own calculation based on the Shannon Evenness Index (SEI)[97]:

$$SEI = -\sum \left( Pi^* \ln(Pi) \right) / \ln(m) \qquad (1)$$

where

$P_i$ = the proportion of coal capacity in each region within a country, using data on regional distribution of coal capacity from ref. 65.

$m$ = the total number of regions within a country. We define regions as administrative subdivisions at the highest level (for example, Zambia is divided into ten provinces, which each include several districts. We use the higher level, province.)

Finally, we include three types of control variables related to the national context: First, variables measuring state capacity: (1) Hanson and Sigman's (HS) index[98] which incorporates indicators of extractive, coercive and administrative dimensions of capacity[98], and has been shown to be a robust predictor of coal phase-out[32]. (2) The Government Effectiveness Index from the World Bank which captures quality of public and civil service, the quality of policy formulation and the commitment of a government to its policies[99]; and "focuses strongly on the administrative aspects of state capacity"[100]. Second, variables measuring economic capacity: (1) The size of the national economy (GDP) for the year in which coal phase-out was pledged converted to USD2020 from the IMF World Economic Outlook[84]. (2) GDP/capita (Purchasing Power Parity (PPP)) for the year in which coal phase-out was pledged converted to USD2020 from the Penn World Table[101]. And third, a variable on access to international funding capturing whether a country is a donor or recipient of either Official Development Assistance (ODA) or EU funds. We code this variable based on data from ref. 102 and a report on EU finances[103].

All models include a measure of the ambition of coal phase-out pledges, since our goal is to test the relationship between ambition and compensation while controlling for other potentially relevant variables. We limit the number of independent variables to a maximum of four in each model due to the relatively small sample (39 countries). To avoid multi-collinearity, we also only use one variable per variable category and exclude any variable combinations with high correlation (Pearson's $R^2 > 0.7$).

We run five sets of multivariable regression analyzes: a central set with our central estimates of compensation and coal phase-out pledge ambition; two sets where we vary ambition using an optimistic and pessimistic interpretation of coal phase-out pledges; and two sets where we vary compensation policies using both an upper and lower estimate for compensation (Supplementary Tables 12-16).

This returns 820 machine-generated models. We find that there are conflicting results on the relationship between economic and state capacity and compensation, where for example poorer countries have higher compensation, when we don't control for access to international funding. We thus pool measures of economic or state capacity with access to international funding, meaning that we include only economic or state capacity as control variables in models which also control for access to international funding.

This results in 330 remaining models: – 66 models from each of the five regression analyzes. We rank the models from each respective analysis according to Akaike's Information Criterion (AIC) (Supplementary Note 3). AIC indicates goodness of fit, penalizing for additional independent variables[104]. A lower AIC means a better model fit. We report our top ten models for all five sets of regression analyzes (Supplementary Tables 12-16).

## Estimating compensation and its uncertainty range for China and India under climate pathways

To estimate compensation for China and India, we use the central average of compensation/ton avoided emissions ($37.5/tCO$_2$), calculated based on all countries with coal phase-out and compensation policies, and the median of avoided emissions in 1.5 °C-, 2 °C- and 2.5 °C-compatible pathways from the IPCC AR6 database for the China+ and India+ regions[86]. We use the China+ and India+ regions to estimate compensation in China and India respectively, since each country accounts for at least 97% of coal power generation in their respective region. In identifying the median among IPCC AR6 pathways, we exclude a handful of pathways (ten for 2 °C and 25 for 2.5 °C) which depict a coal power expansion, and thus higher emissions from coal

power than in the reference scenario as this has been criticized as unrealistic[105].

We calculate the uncertainty ranges for compensation estimates for China and India accounting for three types of uncertainties (Supplementary Note 2): (1) Parametric uncertainties arising from coal phase-out pledges and compensation policies; (2) Model uncertainties arising from using different control variables and the confidence intervals across different regression models; and (3) Pathway uncertainties arising from coal phase-out trajectories for China and India envisioned in different IPCC AR6 pathways and leading to different levels of avoided emissions

We calculate the uncertainty ranges of compensation for China and India using two methods: first, we use the top performing models across our five sets of regression analyzes and their confidence interval (Supplementary Tables 12-16). Second, we use a range of average compensation per ton of avoided emissions based on varying coal phase-out pledge ambition and compensation estimates (Supplementary Table 10 and Supplementary Note 2). We apply both methods to the IQR of avoided emissions in IPCC pathways.

## Comparing compensation to national and international policy support

We also benchmark compensation against several domestic energy and climate policies, international financial support mechanisms, as well as recent coal power plant costs.

For domestic energy and climate policies we consider average carbon price data from 2017 to 2022 under the EU emissions trading scheme from emission spot primary market auction reports[106]; and annual coal production subsidies from the Organisation for Economic Co-operation and Development (OECD)[107] and International Institute for Sustainable Development (IISD)[108] (Supplementary Table 7 and Supplementary Fig. 3). We do not include coal consumption subsidies, since compensation for coal phase-out generally focuses on producers (companies, workers, and regions) rather than consumers; we also do not include investments in state owned enterprises in our main estimate because these are made under the assumption that enterprises are operational and will return a profit, while compensation is paid without an expected return. We could not identify coal subsidy information for Vietnam, Bulgaria, Croatia and Romania.

For international financial mechanisms, we include only public finance which is most comparable to the policy effort for accelerated coal phase-out and since private investment is likely driven by a different logic. We include average annual gross Official Development Assistance (ODA) disbursements over the period 2013-2022 (Aid type: "Memo: ODA Total, Gross Disbursements")[43]; annual climate finance first pledged by developed countries to developing countries at the 15th Conference of the Parties (COP15)[25]; and a climate finance request from India's prime minister for $1 trillion at COP26[109]. To compare this to annual compensation estimates (Fig. 3), we divide the request by the median duration of coal phase-out in line with 1.5 °C- and 2 °C- compatible pathways (Supplementary Table 7).

To compare coal phase-out subsidies and the international financial mechanisms to compensation, we calculate an annual compensation rate by dividing the upper and lower estimates for compensation by the number of years from the announcement of the coal phase-out pledge to the year of planned coal phase-out. For countries with uncertainty in the pledge date (Supplementary Table 3) we use the longer coal phase-out duration for the lower estimate and the shorter duration for the upper estimate, assuming that more ambitious pledges would be accompanied by higher compensation. For annual compensation estimates for China+ and India +, we divide pathway-specific coal compensation by pathway-specific phase-out dates (the first year in which unabated coal power generation declines below 1%).

Finally, for coal power plant costs, we identify all recently-constructed coal plants in the EU between 2010 and 2022 from the S&P database[65] using a systematic Google search with the terms: "[power plant name]" + "construction cost" + "[year of construction]" (Supplementary Table 6). To compare these costs to compensation, we normalize compensation to the installed coal capacity in the year the coal phase-out pledge was made[65].

## Limitations

Our analysis is based on currently evolving coal phase-out pledges and compensation policies. This dataset needs to be updated as new countries develop compensation policies, and more information about the design and implementation of such policies can be collected. Additionally, given the early phase of such policies it is not possible to evaluate their implementation or impact. It will be particularly important to understand how regions, workers and industries supported by compensation fare in the long term and what role compensation plays in this development. As compensation policies evolve along their implementation and evaluation phases, uncertainties around the amount of compensation will decrease – such as current uncertainties around future compensation particularly for JETP countries. Our database and analysis should be updated as more information becomes available. We also do not examine the difference between grants versus loans for international compensation such as the JETPs – future research is needed to understand the conditions under which grants or loans are pledged, and how this affects the implementation of coal phase-out compensation.

Additionally, while the policy documents we review provide some information on how compensation policies are financed, compensation likely originates from additional sources. Where we could not access national policy documents in a language known by the authors, we reached out to country experts (for example for Poland) or retrieved information from international organizations (for example for Greece) to minimize the effect of language barriers on our data collection. However, it is possible that additional resources may be accessible to native speakers for certain countries.

Finally, the predictive power of statistics tends to increase with the number of cases. Currently, a relatively small number of countries have both coal phase-out pledges and associated compensation policies. If more countries are added to the database in the future, the regression analysis should be updated to understand whether the relationship we currently observe between avoided emissions and amount of compensation remains stable.

## Reporting summary

Further information on research design is available in the Nature Portfolio Reporting Summary linked to this article.

## Data availability

The compensation data generated in this study are available on Zenodo at https://doi.org/10.5281/zenodo.10782166. Coal power plant data has been retrieved from the S&P Global World Electric Power Plants Database[65]. Data on IPCC pathways used in this study are available in the IPCC AR6 scenarios database[86]. Data on coal phase-out commitments used in this study are available in ref. 23. Data on coal-based power generation used in this study are available in the IEA World Energy Balances database[92]. Data on amount of coal mined are available in the Enerdata database[94]. Employment factors for coal power generation and coal mining used in this study are available in ref. 96. Data on national GDP and inflation used in this study are available in the IMF World Economic Outlook database[84], and data on GDP per capita adjusted for Purchasing Power Parity used in this study are available in the Penn World Table[98]. Data on government effectiveness used in this study is available in the World Bank's Worldwide governance indicators database[99]. Data on Hanson and Sigman's Index are available in ref. 98. Data on Official Development Assistance is available in the OECD database[43]. Data on coal production subsidies used in this study are from the OECD[107] and the IISD[108]. Source data for the figures and tables in the main text are provided with this paper.

## Code availability

The code for calculating avoided emissions is available online at https://doi.org/10.5281/zenodo.10776566.

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

## Acknowledgements

The research leading to the results reported in this study was supported by the CINTRAN project (84539; L.N. and J.J.) made possible by funding from the European Commission's Horizon 2020 program, by the MAN-IFEST project (950408; V.V. and A.J.), funded by the European Commission's Horizon 2020 ERC Starting Grant programme and by the ENGAGE project (821471; A.C.), funding by the European Commission's Horizon 2020 programme.

## Author contributions

JJ & AC conceived the study. JJ, AC and LN developed the Methodology. LN led data collection and curation. LN, VV, and AJ conducted formal analysis. LN and JJ wrote the original draft with contributions from AC. LN and JJ revised the manuscript with contributions from AC, VV and AJ. LN, JJ, AC, VV and AJ contributed to visualisation. JJ provided supervision and LN project administration. JJ acquired funding.

## Funding

## Competing interests

The authors declare no competing interests.
