## [Peer Review File · Nature Communications]

REVIEWER COMMENTS

Reviewer #1 (Remarks to the Author):

This is a timely and original piece of research, contributing significantly to scholarship and policy debates on coal phaseout, and especially in the context of Just Energy Transition Partnerships. It builds on a growing field of research – by these authors and others – on how to overcome the challenges of coal phaseout. The study analyses compensation payments in countries that have committed to phase out coal – their amounts, sources and beneficiaries – and thereby gives valuable insight into mechanisms and financial amounts for enabling coal phaseout in countries that have not yet committed to it.

The key findings are that:

- The largest coal consumers that have committed to phaseout have established policies to compensate those negatively affected;
- About half of this compensation comes from national and half from domestic sources;
- Based on existing compensation policies, phasing out coal consistent with 1.5°C would cost USD 2.3 trillion in China and 1 trillion in India.

I have three main comments (1-3 below), followed by some modest methodological issues (4) and various minor comments grouped under headings (5-7).

1. Breakdown of beneficiaries

I would really like to see a breakdown of compensation amounts by beneficiary, as this is a key element of understanding compensation to date, and its implications for future compensation policies.

It seems to me that beneficiaries fall broadly into about four categories (workers, regional economies, coal companies, renewables). I would suggest breaking down amounts between types of beneficiary (and “other”, “not specified” etc) in Table 1.

From a quick and rough addition based on Supplementary Table 1 (upper estimates), it looks like about \$60bn for regional economies, \$25bn for coal companies, \$15bn for renewables (primarily Korea) and \$6bn for workers (primarily Germany). Contrary to lines 132-134, workers do not (directly) look like a large part of the beneficiaries.

2. Variables tested in the regression

My biggest concern about the study relates to the regression, which is key to determining those China and India estimates. Whilst the methodology sets out to test variables that may plausibly affect compensation, based on the literature and the understanding of barriers to phaseout, there are several other variables that could have been tested, which I would expect to be relevant.

On page 20, I would say that only variable #1 reflects the ambition of a phaseout policy. Ambition level is especially important: a more ambitious phaseout timeline would likely require more compensation for any given country (especially as most of the committed phaseouts are slower than would be consistent with the Paris goals – below). Variable #1 reflects only the value of early-retired coal plants themselves, which based on my reading of Supp Table 1, is less important than regional economies. The economic diversification literature suggests a lot of time is needed to transform a national or regional economy, and so likely doing it faster will require larger investments (including some that fail). One simple additional ambition-related variable to consider could be the time over which the phaseout takes place.

Variables #2-5 on page 20 are all characteristics of the coal sector, essentially reflecting system inertia. Again, the tested variables focus mainly on the value of coal plants and mines themselves. It would be good to also consider regional concentration of coal power within a country, as this will shape the specific localised costs. A second element is the number of workers affected (this has regional economic impacts as well as direct ones), which SI lines 212-215 reject using due to quality of available data. I would suggest even approximate estimates – perhaps using stylised assumptions to convert available data into consistent ones – here would be useful to test in the regression; after all, there are significant uncertainties in some of the other variables too, and in the regression itself. A third possibility is coal's share of power generation: while the paper is estimating absolute compensation, relative measures of difficulty are relevant as well as absolute ones (e.g. polyarchy, GDP per capita).

Related to government capacity, it seems to me that a central variable is whether the country is an ODA recipient, an EU member, or neither, as those two characteristics give access to international funding. (This may also answer the apparent paradox in lines 249-255 of the SI, as less-wealthy countries – notably Indonesia and Vietnam – have access to more international funding).

One small quibble: I would challenge the suggestion in SI lines 158-161 and 201-203 that more democratic countries will pay less compensation. Surely a more authoritarian government will be more able to ignore harms done to part of the population, driving through policies by fiat.

3. Uncertainties in results

The Limitations section of Methods notes the small number of countries in the sample. The resulting uncertainties could be more clearly highlighted in the findings in the main paper.

In addition, the countries with commitments are not diverse: out of 20 countries in Supplementary Table 2, 16 are EU members and 2 are JETP recipients. The China and India estimates assume they “adopt compensation strategies similar to existing policies”; I would suggest an additional caveat that existing policies may not be very representative of the costs in those countries.

Furthermore, just 3 countries account for 77% of the total compensation.

I believe the IQR for India and China (lines 211-212 and Table 3) relate to the distribution of IPCC scenarios? There are also confidence intervals associated with the regression coefficients, which would make the range wider – I suggest including these in the error bars.

4. Other methodological issues

It is a strange methodological choice (lines 564-565) to retire plants abruptly on the phaseout date, in the estimate of avoided emissions. More realistic would be a gradual phaseout (indeed, that’s what the term implies). A simple choice would be a linear decline from date the phaseout commitment is made; an alternative would be sigmoidal. Given the centrality of avoided emissions in the narrative and results, the paper should at least assess sensitivities to this assumption, and/or justify the choice here.

Why are countries with no compensation policies (page 11 of SI) not included in the graphs and regressions as zero values? This would presumably affect the results significantly?

In assessing India and China compensation, why does the paper use IPCC C1 + C2 scenarios for 1.5°C and C3 + C4 for 2°C? It is more common in the literature to focus on low- and no-overshoot 1.5°C scenarios (C1) and likely below 2°C scenarios (C3), to reflect the meaning of the Paris goals.

5. Framing issues

Lines 12-13, (also 267-268): The article reports findings that compensation policies are “essential” to phaseout. I don’t think this is supported by the present research, though it is prima facie plausible. All this research shows is that the largest and most ambitious coal power phaseouts have involved large compensation policies. A simple intuitive reading of the evidence would refute that finding on grounds that are several smaller countries with phaseout plans but no compensation. If an “essential” finding is to be made, I think the regression would need to be expanded to show how unlikely a phaseout is (in a larger coal consumer?) with zero compensation.

Much of the framing of the Abstract and Introduction is around just transition. However, direct support to workers – which would most obviously fit within just transition definitions – appears to be the smallest part of the amounts involved. Structural investments in regional economies could be included in a broader conception of just transition, but compensation payments to coal/power companies cannot. I suggest either making just transition less central to the framing, and/or explicitly noting that the study considers a broader range of compensation.

To my mind, the comparators make the China and India estimates sound unduly high. “17 times higher than all existing compensation” sounds like a lot, but China and India account for seven times as much coal power generation as the 20 countries in Supp Table 1 (according to BP Statistical Review).

The comparison with 2021 ODA implies that it is like-for-like. I suggest clarifying that ODA is an annual cost, but China/India compensation is not. The phaseout compensation is presumably over about 15 years. It is still high, but not quite as extreme as equality with ODA makes it sound.

6. Suggested additional info that would be good to include (if easily done)

Very few of the phaseout commitments are consistent with the Paris goals. By estimating 1.5°C/2°C-consistent compensation for only China and India, there is an implication that the countries with

commitments have done what is needed, and a suggestion that the primary onus is on China and India (whilst those are indeed the largest coal consumers, they are not the only ones that need to do more). It would be interesting – and presumably not too difficult with the existing methodology? – to look too at what would be needed to upgrade existing commitments to Paris-consistent, and/or achieve Paris-consistent phaseouts in other major consumers that have not agree to phaseout, notably the United States.

Some smaller points:

Table 1: I suggest adding a column stating the phaseout year – this is important because of the earlier mention of 1.5°C and 2°C in the Introduction, and again the risk of implying that these phaseouts are Paris-aligned

Page 7: What is the mean compensation per tCO₂? This would be a useful finding.

Page 7 and figure 2: For national versus international shares of compensation, it would be interesting to separate donor from ODA-recipient countries (or annex 1 from annex 2, or EU+OECD from ROW etc).

Since the regression is central to estimating the China and India compensation requirements (if I understand the methodology correctly), I suggest making the regression results clearer in the main text, rather than just in Methods (lines 637-641).

7. Clarity issues

Line 13: The finding of proportionality to avoided emissions should probably add “when controlling for XXX”, as otherwise it looks like a conflict with Figure 2.

Lines 80-82: It would be clearer to say “We define this as the difference in cumulative emissions between two scenarios:...”

Table 1: What does “Support closure of coal plants” mean?

Table 1: Is there a difference between “assisting laid-off workers” and “support for laid-off workers”?

Table 3: For the four individual countries, the range of phaseout year refers to an uncertainty range, whereas for “all other countries” it reflects a range among actual data points. I find it confusing to give a single year of 2036, which is presumably a mean or a median; better just to give the range, or express it differently, to show that it means something different from the other four rows. It would also be nice to separate “all other countries” between Global North and Global South.

Line 231: “would require using most...” implies that the amount of climate finance is fixed. I suggest changing to “would require an amount equal to...”

Lines 179-180 (also 552-553): The statement that avoided emissions are proportional to stranded capacity is not clear to me. Surely utilisation rates vary between countries? That said, it doesn’t need to be proportional for the argument being made here: it is enough to say that retired capacity and avoided emissions are both appropriate measures of phaseout ambition.

In those same lines, I would suggest avoiding the word “stranded”, which is generally taken to refer to investments that make a less-than-expected or less-than-commercial return. Better would be “retired” or “retired early”.

Lines 554 to 563 (and possibly also 80-81): It would be good to note that the reference scenario assumes no new coal plants are built (if I understand correctly). This is a fair assumption, although in reality coal plants are still being added in some countries, as the paper notes (lines 193-194).

Lines 557 to 559: I am confused by this sentence. Since this paragraph is about the reference scenario, where retirements follow their natural course, why are pledges having an effect?

Supplementary Table 2: The heading of column 5 appears to have been truncated. I first thought it might be number of operating coal plants but the numbers are too high for that (e.g. Germany has 60, Vietnam 25, according to Global Energy Monitor); is it individual units of power plants? As of what date?

Reviewer #2 (Remarks to the Author):

The article provides an updated analysis of compensation levels directed at regions and actors affected by coal phase-out policies in countries with coal phase-out targets. A database is developed to perform an analysis of the effect of compensation plans for the realization of ambitious coal-phase out targets that are related to the Paris Agreement's objectives (1.5-2°C). The authors conclude that compensation is crucial for the feasibility of ambitious coal phase-out policies. Hence, the underlying puzzle that the article attempts to address seems to be the tensions between tempo and costs in the energy transition, i.e. how much will it cost to speed up the transition away from coal.

My main concern is that the article confines itself to considering compensation funding as the only just transition policy measure worth investigating. Given the article's reliance on quantitative methods, it is to a certain degree understandable that the authors need numbers to measure the effects that they are interested in. However, previous studies that discuss the tensions between various concerns in energy transitions – such as costs, tempo, and justice/fairness – point to several additional factors beside compensation as crucial for the political feasibility of fossil fuel phase-out, including distributional justice, procedural justice and restorative justice (McCauley & Heffron, 2018; Trencher et al. 2020; Gürtler et al., 2021; Healy & Barry, 2017; Isoaho & Markard, 2020; Leipprand & Flachsland, 2018; Rentier et al., 2019; Harrahill & Douglas, 2019; Green and Gambhir 2020). While not suggesting to upend the current article completely, I would like to see more reference to these important contributions to the just transition literature when the authors discuss why they have chosen to confine their analysis to compensation measures only. The article should relate compensation measures to the policy priorities arising from distributional justice, procedural justice and restorative justice concerns in just transition policy processes, for instance explaining why re-skilling or public investments in R&D are important compensation-relevant policy measures in affected regions in some countries.

Another concern, related to the above, is that the results provided from the article's analysis is quite unsurprising. An overview of compensation levels cross-checked with coal phase-out targets can give us a cursory insight into how much money policymakers in each country are willing to spend, but unfortunately produces no insights into why and how coal phase-out policies has come about in individual countries. Specifically, the analysis is not able to answer the pertinent concern that the authors themselves raise, namely that the socio-political feasibility of rapid coal phase-out is unclear. More attention to the key dimensions of just transition policies (distributional, procedural, restorative) as mentioned above will allow the authors to explore their findings in a more nuanced way. Moreover, meaningful comparison across countries is also restricted with the current analytical set-up because no attention is given to varieties of state capacities (Meckling and Nahm 2021) or varieties of capitalism

(Hall and Soskice 2001) beyond the quite shallow democracy/non-democracy distinction included. My recommendation is that the authors include reference to such crucial differences in the policymaking processes across the countries included in the analysis, perhaps finding a way to measure differences (or discuss more qualitatively/empirically) in a way that could add substantially to the insights into why countries have different approaches to coal-phase out policies.

I hope these comments can be useful for the authors in the review process!

Reviewer #3 (Remarks to the Author):

This study offers an important and original contribution to the rapidly increasing literature on coal phase-outs. The study points to a set of clear conclusions that appear well supported by the data. The conclusions also have important implications. I believe that the following comments could help improve the manuscript.

Title: You might consider adding a hint about what you found in the this paper. You might also considering answering: Is the socio-technical cost affordable, feasible etc? Is more effort required?

Introduction

The authors do a good job at summarizing the situation studied.

Financial compensation packages: The authors do not spend many words defining that such packages might entail. Since they would take many shapes and forms, a few more words should be devoted to unpacking this generic approach so that novices might understand.

A clearer statement about the research gap should be given. The first sentence of the methods (Building a database...) could be moved to the introduction and expanded. See the type on page 2 ("I")

Page 2: The objective of the method “To evaluate the intensity and speed of national coal phase-out commitments” should be made clearer, as I had to read well into the paper before I saw how this connected to your main theme.

Findings: These are clear and easy to follow.

Page 7-8: We test the possibility of predicting compensation .. I was unsure of the meaning of this statement. I had to check the supplementary material to see that you conduct a regression to predict the size of the compensation and not just the presence of compensation packages. This should be made clearer in the main text.

Gt of avoided emissions in tables. It should be made clear if this is carbon or CO₂ (or CO₂e).

Table 3. I was unsure to how interpret this analysis because you start this section with a focus on India and China but include Indonesia and Vietnam. The logic for including these other countries in Table 3 should be explained in the note.

Conclusion: After reading your paper, when I arrived at the conclusion I was unsure of the evidence that supports the argument “Our research suggests that compensating affected actors is necessary for accelerated coal phase-out” and “By comparing coal phase-out in countries which have compensation plans with those that don’t, we show that compensation policies are essential to realizing premature retirement of coal (abstract). I had to re-read the paper to find the evidence and found it as follows:

“We find that all countries with ambitious coal phase-out commitments also plan compensation for coal phase-out (Figure 1). The idea here is that countries with compensation plans are able to realise the largest avoided emissions relative to countries that have a phase-out plan but don’t have compensation plans. This is an extremely important finding and a few more words should be devoted to explaining the figure and making sure that the reader does not miss this finding and the evidence/logic underpinning this.

Dear Reviewers,

Thank you for your thoughtful and detailed comments which have helped us to strengthen our manuscript. We have significantly expanded our regression and uncertainty analysis, added analysis on beneficiaries of compensation, estimated compensation for China and India for a less ambitious climate target, and revised the structure and framing of our work to clarify our contribution and better connect to the just transitions literature.

Regarding the regression, we added 7 new independent variables, expanded case selection to include all countries with coal phase-out pledges (including those without compensation), and replicated our regression with an uncertainty range for phase-out pledges and compensation policies. We also triangulated our estimates of compensation for China and India using the average compensation per ton avoided CO₂ emissions implied by existing policies.

These revisions significantly increased the robustness of our results but did not change our conclusions. The number of cases increased from 21 to 39 and number of regression models we test from nine to 820. We still find that avoided emissions is the best predictor of compensation for coal phase-out including when we replicate our regression analysis using optimistic and pessimistic interpretations of coal phase-out pledges as well as upper and lower estimates of compensation (Supplementary Tables 11-15).

Our resubmission also captures a wider range of uncertainties. In our prior submission, we estimated the uncertainty of compensation for China and India to phase-out coal in-line with climate targets using the IQR of avoided emissions from coal phase-out in IPCC pathways. Now, we also consider parametric uncertainties arising from existing pledges and compensation policies as well as model uncertainties from different control variables and confidence intervals for regression models (Supplementary Table 11-15, Figure 4). While the uncertainty range is wider, our estimates are consistent with our prior submission (see table).

Table 1. Compensation estimates for China and India (trn USD) in prior and re-submission

	1.5°C TARGET		2°C TARGET	
	prior submission	re-submission	prior submission	re-submission
CHINA	\$2.3 [1.7-2.6]	\$2.2 [1.2-5.3]	\$1.7 [1.5-2.2]	\$1.6 [1.0-4.8]
INDIA	\$1 [0.8-1.1]	\$1 [0.5-1.6]	\$0.8 [0.6-0.9]	\$0.8 [0.5-1.3]

We also expanded our analysis of beneficiaries (Figure 2, Supplementary Table 2) and added an estimate of compensation for China and India consistent with a 2.5°C pathway which based on previous literature[refs] would be consistent with the pace and intensity of coal phase-out pledges of large coal consumers (Supplementary Figure 5).

Finally, we significantly revised the framing and structure of our manuscript to more clearly articulate our contribution and its connection to the just transition literature. We also restructured the results to more clearly highlight our main findings as well as our use of multiple regression analysis. We reworked the Discussion to reflect how our findings relate to different conceptions of justice (which we expand in Supplementary Note 4).

We thank the Reviewers once again for their time and look forward to your further comments. Below you find a point-by-point explanation for how we responded to your comments. Our responses are in blue and quotes from the main text in green.

Reviewer #1 (Remarks to the Author):

This is a timely and original piece of research, contributing significantly to scholarship and policy debates on coal phaseout, and especially in the context of Just Energy Transition Partnerships. It builds on a growing field of research – by these authors and others – on how to overcome the challenges of coal phaseout. The study analyses compensation payments in countries that have committed to phase out coal – their amounts, sources and beneficiaries – and thereby gives valuable insight into mechanisms and financial amounts for enabling coal phaseout in countries that have not yet committed to it.

The key findings are that:

- The largest coal consumers that have committed to phaseout have established policies to compensate those negatively affected;
- About half of this compensation comes from national and half from domestic sources;
- Based on existing compensation policies, phasing out coal consistent with 1.5°C would cost USD 2.3 trillion in China and 1 trillion in India.

We thank the reviewer for their positive reception of our work as well as for their detailed and constructive reviewer comments that we found extremely helpful.

I have three main comments (1-3 below), followed by some modest methodological issues (4) and various minor comments grouped under headings (5-7).

Thank you for structuring the comments in this way. For comments which contain several ideas and for which we made several revisions, we split the comment and our response for clarity.

1. Breakdown of beneficiaries

I would really like to see a breakdown of compensation amounts by beneficiary, as this is a key element of understanding compensation to date, and its implications for future compensation policies. It seems to me that beneficiaries fall broadly into about four categories (workers, regional economies, coal companies, renewables). I would suggest breaking down amounts between types of beneficiary (and “other”, “not specified” etc) in Table 1.

We thank the reviewer for this suggestion, and agree that a breakdown of beneficiaries helps to better understand the nature of compensation policies across different countries. We added Figure 2 to the main text where we show the distribution of compensation between different types of support: for regional development to authorities and to SMEs respectively, for unemployment benefits and re-training, for renewables capacity and low-carbon infrastructure, and for coal power plant and mine closures.

We also added Supplementary Table 2 where we describe the different types of support in more detail. We have also expanded Supplementary Table 1 to cover the type of support which each individual funding mechanism relates to as well as how we coded specific funding mechanism in the same table.

Note that we have chosen to use the terms “scope” and “type of support” rather than “beneficiary” for each funding mechanism because it was not always possible to identify which actor will receive funding under a specific mechanism. For example, for Bulgaria, we identified compensation to “design, build and commission infrastructure adequate for transmission of hydrogen and low-carbon gaseous fuels”¹, which specifies the ultimate purpose of compensation, but does not specify the recipient (this funding could be either directed to companies or regional authorities to commission hydrogen infrastructure). Additionally, we identify several cases in which regional authorities receive funding to support renewables infrastructure (while in other cases, energy companies receive funding for the same type of support), or to support retraining options for former coal workers (while in other cases, workers receive funding directly).

From a quick and rough addition based on Supplementary Table 1 (upper estimates), it looks like about \$60bn for regional economies, \$25bn for coal companies, \$15bn for renewables (primarily Korea) and \$6bn for workers (primarily Germany). Contrary to lines 132-134, workers do not (directly) look like a large part of the beneficiaries.

The reviewer is right. In countries where information is available, the largest financial flows go to regional economies and companies (Figure 2). We have rewritten the text to reflect this:

Aside from JETP funding, most compensation is earmarked for regional development in coal dependent regions (central estimate \$66 billion or 61% of all non-JETP funding cases – Figure 2, Supplementary Table 2). The second largest amount goes to coal mining and power plant companies, across six countries (central estimate \$23 billion or 21% of all non-JETP funding). Another \$14 billion (central estimate), or 13% of all non-JETP funding, goes to renewable capacity and low-carbon infrastructure expansion channeled through renewable companies and regional authorities (most notably in South Korea where all funding is earmarked for this purpose). Only four countries include support for unemployment benefits and worker retraining and overall this represents one of the smallest amounts of funding (\$6 billion in total). Finally, a small portion of regional funding in one case is paid to local small and medium sized enterprises to support the diversification of the regional economy (\$0.2 billion). For three countries (Ukraine, Chile, and North Macedonia), we were able to identify the type of support associated with compensation, but could not quantify the overall level (Table 1).

We should also note, however, that the largest financial flows are allocated to national governments through JETPs (\$88 bln central estimate). The distribution between different types of support has not been announced yet, so we are not able to quantify this amount for the figure on different types of support (Figure 2) but in Table 1 we describe the types of support which are mentioned in JETP agreements.

2. Variables tested in the regression

My biggest concern about the study relates to the regression, which is key to determining those China and India estimates. Whilst the methodology sets out to test variables that may plausibly affect compensation, based on the literature and the understanding of barriers to

phaseout, there are several other variables that could have been tested, which I would expect to be relevant.

We thank the reviewer for their comments on the regression analysis and the recommendations for further variables to include. We have substantially expanded the regression analysis as well as modified how we discuss the results in the main text and SI in response to this comment and other reviewer comments related to the regression analysis. Here we provide an overview for the Reviewer of the main revisions to the regression analysis. In comments below, we explain how we have incorporated each of the specific suggestions related to the regression analysis.

First, we added eight additional variables in the regression analysis:

- time over which phase-out takes place;
- coal share of power generation;
- regional concentration of coal power plants;
- number of workers coal workers;
- whether a country is a donor or recipient of international funding;
- World Bank Government Effectiveness indicator; and
- Hanson-Sigman index (a measure of state capacity).

In the majority of cases, these variables were added to our analysis but in two cases we used one of the new variables to replace a variable in our previous submission. The coal jobs variable replaced the value-added of the coal sector. This is because in this submission we expanded our set of cases as suggested by the Reviewer (see page 12) and we could not find data of the value-added of the coal sector for these new countries. Second we replaced the polyarchy variable with the World Bank Government Effectiveness indicator and the Hanson-Sigman index since these variables have been identified in the literature as better predicting coal phase-out and both Reviewer 1 and 2 raised concerns with the polyarchy variable (see more discussion on pages 7-8).

We thus now use 13 variables compared to our first submission which used eight variables. We elaborate on each of the new variables in our answers below, in the Methods section, and in Supplementary Note 3.

To deal with increasing complexity and risk of co-linearity from additional variables², we follow previous approaches in the literature and assign a specific causal mechanism to each variable (e.g. ref³). Our control variables fall under six broader mechanisms arising from characteristics of the coal sector and characteristics of the national context, as well as the ambition of the coal phase-out (Methods and Supplementary Note 3). In each model, we include no more than one variable for each mechanism.

We also replicated our regression analysis using both an optimistic and pessimistic interpretation of coal phase-out pledges to capture the range of plausible avoided emissions from coal phase-out (our key independent variable) and for upper and lower compensation estimates (our key dependent variable). Consistent with our previous submission, we find that avoided emissions is the best predictor of the cost of compensation policies, statistically significant at the highest level in our 10 best-performing models in each of the five sets of regression analyses (Supplementary Tables 11-15).

Nevertheless, we understand the Reviewer’s reservations on using the regression analysis to estimate the compensation needed for China and India given our limited number of cases. In this submission, we have triangulated this calculation by using the average compensation per ton of avoided CO₂ emissions implied by existing compensation policies (Figure 1, Supplementary Note 2).

In our previous submission, we calculated the central estimate of compensation for China and India to meet climate targets using the top regression model. In our resubmission we triangulate compensation estimates for China and India using results from our ten best-performing models for each set of our regression analyses as well as the average compensation per ton avoided emissions (50 models in all). We find that the range of compensation per ton avoided CO₂ emissions in our top-performing regression models (\$29-46) is consistent with the average compensation per ton of avoided emissions \$27-45. This means that our estimates of compensation are robust to controlling for the effect of confounding variables (relating to the strength of the coal sector and characteristics of the national context) as we do in our regression analysis.

These revisions increase the robustness of our results but do not change our conclusions (see table on page 1).

To summarize, we control for several additional mechanisms which may influence compensation and show that our results are robust not only with our top regression model but our top ten in each of our five sets of regression runs. Additionally, we show that triangulating our results from regression models using an alternative method (using the average compensation per tCO₂ of avoided emissions) yields consistent results with our regression-based results.

On page 20, I would say that only variable #1 reflects the ambition of a phaseout policy. Ambition level is especially important: a more ambitious phaseout timeline would likely require more compensation for any given country (especially as most of the committed phaseouts are slower than would be consistent with the Paris goals – below). Variable #1 reflects only the value of early-retired coal plants themselves, which based on my reading of Supp Table 1, is less important than regional economies. The economic diversification literature suggests a lot of time is needed to transform a national or regional economy, and so likely doing it faster will require larger investments (including some that fail). One simple additional ambition-related variable to consider could be the time over which the phaseout takes place.

We thank the Reviewer for this suggestion. Indeed, the ambition of a phase-out policy is very important in shaping compensation. Following the reviewer’s suggestion, we tested the time over which coal phase-out takes place (calculated as the number of years between the year when the coal phase-out pledge is made and the year of planned coal phase-out). This variable does not feature in our ten best-performing model (the first model it features in is the 15th model (ranked by AIC) in the regression based on pessimistic avoided emissions, and not found to be significant).

Our hypothesis is that the lack of a strong effect for the “length of phase-out” variable is that some countries phase-out coal quickly but the fleet they are phasing out is very small

and has very little regional development impacts. For example, Austria has a very fast phase-out (three years) but a very small coal fleet (0.8 GW of installed coal power) and no coal mining. In contrast, a country like Germany has a longer phase-out period (11 years) but a larger coal fleet and phases out domestic mining.

Avoided emissions strikes a balance between capturing the size of the coal fleet and the speed of retiring coal plants prematurely.

Variables #2-5 on page 20 are all characteristics of the coal sector, essentially reflecting system inertia. Again, the tested variables focus mainly on the value of coal plants and mines themselves. It would be good to also consider regional concentration of coal power within a country, as this will shape the specific localised costs.

Many thanks for highlighting this important dynamic. We added a variable where we calculate regional concentration of coal power within each country using Shannon's evenness index⁴ which measures diversity between a value of 0 to 1. The closer to 0, the less diverse i.e. the more regionally concentrated is the national coal power plant fleet. For example, if all power plants are located within one region, the Shannon Evenness Index is at 0. Seventeen of the best-performing models across our five sets of regression analyses control for regional concentration of coal power.

A second element is the number of workers affected (this has regional economic impacts as well as direct ones), which SI lines 212-215 reject using due to quality of available data. I would suggest even approximate estimates – perhaps using stylised assumptions to convert available data into consistent ones – here would be useful to test in the regression; after all, there are significant uncertainties in some of the other variables too, and in the regression itself.

We agree with the reviewer that the number of affected workers may be an important indicator of socio-political barriers to phasing out coal. In the first submission, we had attempted to collect empirical data on coal workers, but were not able to do so for all countries in our sample. In this revision, we follow the Reviewer's suggestion and design an indicator of coal jobs using the national job intensities from ref⁵ which report jobs per MW of installed coal capacity, MW of capacity under construction, and PJ-equivalent of domestic coal production. We used these job intensity values together national installed coal capacity, coal capacity under construction, and amount of domestic coal mining, to estimate the number of coal workers per country.

For those countries where we could identify empirical data on workers, we also validated the empirical data with our estimates using job intensities from ref⁵ and found very good agreement (correlation >0.9).

Eighteen of the ten best-performing models across our five sets of regression analyses include amount of coal workers, almost all at the highest level of statistical significance.

A third possibility is coal's share of power generation: while the paper is estimating absolute compensation, relative measures of difficulty are relevant as well as absolute ones (e.g. polyarchy, GDP per capita).

We have also added coal share as a variable. We calculate coal as a share of the national electricity system. However, we find that coal share is not a robust control variable in our model: we would expect coal share to be positively correlated to compensation cost, however the model returns negative co-efficients for this variable. We think that the coal share variable is interacting with the effect of total coal capacity, which is captured by avoided emissions, and is thus shown as negatively correlated in the model. We have thus decided to exclude all models which contain coal share .

Related to government capacity, it seems to me that a central variable is whether the country is an ODA recipient, an EU member, or neither, as those two characteristics give access to international funding. (This may also answer the apparent paradox in lines 249-255 of the SI, as less-wealthy countries – notably Indonesia and Vietnam – have access to more international funding).

We agree with the Reviewer that the availability of international funding may be an important factor shaping compensation since access to funding could enable higher compensation and could also explain why less wealthy countries have higher compensation. In our updated regression analysis, we added a binary variable where we separate countries into two groups:

- the first group includes countries that are eligible for Official Development Assistance (ODA) and countries that are net-recipients of EU funding (i.e., they receive more transfers from the EU than they pay into the EU budget);
- the second group includes countries that are neither ODA-recipients nor EU-recipients.

41 of the ten best-performing models across our five sets of regression analyses (ranked by model fit) control for this variable, with a positive correlation to the amount of compensation thus supporting the hypothesis that access to international funding enables higher compensation.

As the Reviewer suggested, we find that this also resolves the apparent paradox in our first submission of less-wealthy countries seemingly having higher compensation. In fact, we find that when we control for access to international funding as we do in this submission, we no longer see this result. For this reason, we now exclude models that only control for economic or state capacity without controlling for access to international funding.

One small quibble: I would challenge the suggestion in SI lines 158-161 and 201-203 that more democratic countries will pay less compensation. Surely a more authoritarian government will be more able to ignore harms done to part of the population, driving through policies by fiat.

Many thanks for raising this issue. We understand the Reviewer’s concern that the negative correlation of the level of democracy with the amount of compensation is counterintuitive. The relationship between the level of democracy and coal phase-out may operate in different ways. On the one hand, as the Reviewer notes authoritarian governments may be able to push policies through that harm parts of the population. At the same time, democracies may be better able at balancing interests of the broader population (which may support coal phase-out) against resistance from vested interests of the coal industry. To use refs⁶ terminology, the selectorate within democracies is much bigger and broader

than the selectorate of autocracies⁶ and if coal interests are part of the selectorate, then the state is forced to pay them off to achieve their policy goals (if it involves coal phase-out). In contrast, in democracies a much broader population is part of the selectorate which can be used to balance out the coal interests.

Given that the polyarchy variable cannot distinguish between these two mechanisms, we have now excluded this variable. Instead, we added two variables capturing state capacity (Sigman and Hanson's index for state capacities⁷ and the Government effectiveness variable from the World Bank⁸) which previous literature has found to influence national coal phase-out^{9,10}. Indeed, we find that these variables are positively correlated to the amount of compensation.

3. Uncertainties in results

The Limitations section of Methods notes the small number of countries in the sample. The resulting uncertainties could be more clearly highlighted in the findings in the main paper.

We agree with the reviewer that uncertainties are important, and significant for a regression analysis particularly when based on few cases. In response to another reviewer comment (p. 12), we have now added countries without compensation policies to our regression analysis, almost doubling our number of cases. This means that our analysis covers about a third of countries with coal power.

Since overall, our number of cases (39) is still relatively low, we have also highlighted in the Main Text the limited number of cases:

Both estimates [for compensation per ton avoided CO₂ emissions] should be treated with a degree of caution given a relatively limited number of cases in our regression analysis (N=39), that nevertheless cover a third of all countries using coal power.

We have also significantly expanded our uncertainty analysis (Supplementary Note 2) particularly in terms of our regression analysis, which is affected by the limited number of cases.

First, to ensure the robustness of our conclusion that coal phase-out pledge ambition is the best predictor of compensation levels, we replicated our regression analysis to cover uncertainties in pledge ambition and compensation policies (Supplementary Tables 11-15 regression). In our previous submission we ran a single regression analysis with the central estimates for avoided emissions from coal phase-out pledges and the central estimates for compensation policies. Now, we run four additional sets of regression analyses using an uncertainty range for coal phase-out pledge ambition and compensation policies. We consistently find that across all sets of regression analyses in all of the top ten models (50 in all), avoided emissions is the best predictor of compensation significant at $p < 0.1\%$ level.

We used these additional sets of regression analyses to better capture the uncertainty range for compensation estimates for China and India (Supplementary Table 10) – which is the second conclusion we draw using the regression analysis. This means that we are able to capture both the parametric uncertainty associated with coal phase-out pledges and

compensation policies along with the model uncertainty associated with using different control variables and confidence intervals (see more below on page 10).

Additionally, we triangulated our estimates of compensation for China and India using average compensation per ton of avoided CO₂ emissions implied by existing compensation policies with the regression results (Supplementary Note 2, Supplementary Table 9).

In addition, the countries with commitments are not diverse: out of 20 countries in Supplementary Table 2, 16 are EU members and 2 are JETP recipients. The China and India estimates assume they “adopt compensation strategies similar to existing policies”; I would suggest an additional caveat that existing policies may not be very representative of the costs in those countries. Furthermore, just 3 countries account for 77% of the total compensation.

We agree with the important caveat that existing compensation policies may not be representative of actual costs of coal phase-out, and have added this to our discussion and conclusion section:

Two particularly pressing policy questions as just transition policies shift from formulation to implementation is how to ensure good governance of such policies and to what extent compensation covers the “real” costs for different actors, including under the case of international compensation paid as grants versus as loans (Supplementary Note 1).

It is also a very good question to what extent our sample can serve as a good analogue for compensation in China and India. While most countries are EU and OECD countries, some of these countries are large coal consumers that also have coal mining, and as such may be relevant to China and India. We also believe that the cases of Vietnam and Indonesia may be particularly relevant to China and India: these are emerging economies as well as major coal consumers that receive international support for coal phase-out. We have added a sentence to the manuscript on the relevance of especially Indonesia and Vietnam to the cases of China and India:

The two JETP-recipients that can be considered the closest analogues to China and India are Indonesia and Vietnam: both are major coal consumers and have emerging economies.

Finally, despite differences in the context between China and India on the one hand and EU and OECD countries on the other hand, there is no a-priori evidence to suggest that mechanisms affecting the outcomes of political processes differ significantly between these contexts. In fact, there are discussions on a just transition for coal regions and workers ongoing in India¹¹, literature calling on transferring just transition approaches to China¹², and in China a precedent of compensation for steelworkers from significant industry downsizing^{13,14}. We include references to this discussion in a new supplementary note on just transitions (Supplementary Note 4) and have expanded our reference to past compensation to steel workers in China:

While there is a precedent of China mobilizing domestic funding to support re-employment of workers in heavy industries who lost jobs due to overcapacity, the overall

cost was roughly one fifth of the potential annual amount of compensation which we calculate as needed for 2°C pathways.

These ongoing discussions and past compensation to steel workers in China^{13,14}, suggests that phase-out and related compensation is likely to follow a similar logic in China and India as it has followed in other countries.

Regarding the point related to the distribution of compensation between our case countries, while it is true that Germany, Indonesia and Vietnam account for a majority of total compensation, they also account for almost half of avoided emissions (roughly 45%) of all 23 countries with compensation in our sample.

I believe the IQR for India and China (lines 211-212 and Table 3) relate to the distribution of IPCC scenarios? There are also confidence intervals associated with the regression coefficients, which would make the range wider – I suggest including these in the error bars.

We thank the reviewer for this suggestion. Indeed in our first submission, the uncertainty range for compensation estimates for China and India were based on the interquartile (IQR) range of avoided emissions across IPCC scenarios together with the top regression model.

We have revised our calculations for the uncertainty range for compensation estimates for China and India. We describe this new approach in the Methods and Supplementary Note 2 and summarize our revision below.

In addition to pathway uncertainties, the new uncertainty ranges capture two additional types of uncertainties:

- **parametric uncertainties** arising from coal phase-out pledges and compensation policies. We captured this uncertainty by calculating how the average compensation per ton CO₂ emission implied by compensation policies would change based on an optimistic and pessimistic interpretation of the coal phase-out pledges as well as upper and lower estimates for compensation policies (Supplementary Table 9). Additionally, we replicated our regression analysis by calculating four additional sets of regression models where we vary the pledge ambition or the compensation policy (Supplementary Tables 11-15).
- model uncertainties which we capture by using different control variables in different models and the confidence intervals associated with specific regression co-efficients.

To translate these uncertainties into our compensation estimates in China and India, we use the interquartile range (IQR) of avoided CO₂ emissions depicted in IPCC pathways together with both: (1) a projection based on the average range of compensation per ton of avoided CO₂ emissions implied by current compensation policies and (2) a projection using our top regression models (50 in all – Supplementary Tables 11-15).

This new approach has made the uncertainty range significantly wider than our previous submission (see Table 1 on page 1 of Reviewer responses).

4. Other methodological issues

It is a strange methodological choice (lines 564-565) to retire plants abruptly on the phaseout date, in the estimate of avoided emissions. More realistic would be a gradual phaseout (indeed, that's what the term implies). A simple choice would be a linear decline from date the phaseout commitment is made; an alternative would be sigmoidal.

We thank the reviewer for this important question. How to represent the specific coal phase-out trajectories of individual countries is a very interesting question. In our modelling we followed the announced coal phase-out plan in Germany, however in most cases it was unfortunately not possible to identify a national coal phase-out trajectory. We thus relied on both mechanistic reasoning and empirical evidence to formulate our approach.

Our starting point is an understanding of how a coal phase-out pledge is implemented. Generally, after a government pledges coal phase-out it passes national regulations to enact this pledge and then it is power companies and coal plant operators which implement the phase-out in order to follow national regulations. We believe it would be unlikely that power corporations would retire coal power plants before their legally-defined coal phase-out date, and before they reach their "end-of-life". We also don't see evidence that national regulations aim for such a development. For example, countries like Finland ban coal power generation from their coal phase-out pledge date onwards¹⁵ and in Canada, there is an emissions cap for coal power generation from a certain date, essentially banning coal power generation without CCS¹⁶. In other cases, such as Poland, coal phase-out has been announced for the future but legally binding regulations have yet to be introduced¹⁷.

For this reason, we assume that all coal power plants are retired as soon as they reach their average historical national retirement age, or latest at the final coal phase-out date, since another method might overestimate avoided emissions from coal phase-out. This is also generally consistent with Germany's coal phase-out plan which generally retires coal power plants as they reach retirement.

Given the centrality of avoided emissions in the narrative and results, the paper should at least assess sensitivities to this assumption, and/or justify the choice here.

We thank the reviewer for their relevant comment, in response to this, we ran two additional regression models using both an optimistic and pessimistic interpretation of national coal phase-out pledges. For example, under the Global Coal to Clean Power Transition Statement, countries pledge to phase-out coal "in the 2040s" - so our central case for coal phase-out is 2045, and we test for a range of coal phase-out dates, earliest in 2040, and latest in 2049. Our results remain robust in that avoided emissions remains the most significant predictor of compensation cost in our best-performing models (Supplementary Table 11-15)

We include the results from the sensitivity of these additional regression runs in our uncertainty analysis together with the sensitivity of the average of compensation implied by coal phase-out pledges (Supplementary Table 10, Figure 3)

We also test for an alternative measure of the ambition of coal phase-out pledges: the number of years over which coal phase-out takes place. The number of years over which

coal phase-out takes place first features in our 15th model (ranked by AIC) from the regression analysis using pessimistic pledge ambition and is not found significant.

Why are countries with no compensation policies (page 11 of SI) not included in the graphs and regressions as zero values? This would presumably affect the results significantly?

Many thanks for this important question. We have revised our regression analysis to include all countries with coal phase-out pledges including those without compensation policies (for these countries we set the compensation to be 0). In our revised manuscript, we report the results for the new regression analysis where we have made additional changes by including additional variables as suggested. We also include countries with coal phase-out pledges but no compensation in Figure 1 (panel a) and in the table of coal phase-out pledges (Supplementary Table 3).

As mentioned in our response to an earlier reviewer comment ([pages 3-5), these revisions did not change our conclusions. Avoided emissions remains the most consistent predictor of compensation cost when we control for additional variables and is present at the highest level of significance in our ten best-performing models across all five sets of regression analysis. Our estimates for compensation cost in China and India are also consistent with our previous submission (See Table 1 on page 1 of this document).

In assessing India and China compensation, why does the paper use IPCC C1 + C2 scenarios for 1.5°C and C3 + C4 for 2°C? It is more common in the literature to focus on low- and no-overshoot 1.5°C scenarios (C1) and likely below 2°C scenarios (C3), to reflect the meaning of the Paris goals.

Thank you for this question. We interpret AR6-scenarios in line with ref¹⁸ which includes pathways which return to 1.5C after a high overshoot (what AR6 defines as C2 pathways) under 1.5°C-consistent scenarios. Ref.¹⁸ includes “Higher-2C” pathways under 2°C-consistent scenarios, which corresponds to C4 pathways.

We have tested the effect of different scenario categories on our estimates of avoided emissions and amount of compensation, and find that using different scenario categories does not lead to a significant difference in our estimates (see new Supplementary Table 8).

5. Framing issues

Lines 12-13, (also 267-268): The article reports findings that compensation policies are “essential” to phaseout. I don’t think this is supported by the present research, though it is prima facie plausible. All this research shows is that the largest and most ambitious coal power phaseouts have involved large compensation policies. A simple intuitive reading of the evidence would refute that finding on grounds that are several smaller countries with phaseout plans but no compensation. If an “essential” finding is to be made, I think the regression would need to be expanded to show how unlikely a phaseout is (in a larger coal consumer?) with zero compensation.

Many thanks for this important comment. Indeed there are 20 countries with small fleets and unambitious plans that phase out coal without compensation (Figure 1, Supplementary Table 3). We have added these countries to the regression analysis, setting the

compensation for these observations to 0 (see also our response above on page 12). Our results remain robust in that avoided emissions remains the most significant predictor across our best performing models (ranked by model fit), and in that the cost of compensation per ton of CO₂ is comparable to our previous submission (coefficients for avoided emissions ranged from \$34-37, whereas now they range from \$27-45).

Regarding the Reviewer's suggestion to expand the regression analysis to calculate the probability of a phase-out commitment in a large coal consumer without compensation, unfortunately this was not possible because there are no empirical cases for this. Nevertheless, we have restructured and revised the text to clarify this point, for example by adding a section to our results entitled "All countries with politically-accelerated coal phase-out and large coal fleets have compensation policies" (see also pages 27-28 in response to R3's comment related to this point).

Much of the framing of the Abstract and Introduction is around just transition. However, direct support to workers – which would most obviously fit within just transition definitions – appears to be the smallest part of the amounts involved. Structural investments in regional economies could be included in a broader conception of just transition, but compensation payments to coal/power companies cannot. I suggest either making just transition less central to the framing, and/or explicitly noting that the study considers a broader range of compensation.

We thank the Reviewer for raising this important point. The Reviewer raises a good question whether investing in regional economies and compensating companies can actually be defined as part of a just transition. We have added a discussion of controversies in the just transition literature around who should benefit from such policies such as the one raised by the Reviewer, and also added Supplementary Note 4 where we discuss different conceptualizations of the Just Transition concept and how different beneficiaries are considered in this discussion.

Compensation policies can be justified not only on the basis of their political expediency making transitions feasible, but also on ethical grounds of ensuring 'just transitions'. However, an opposite position also exists which views compensation policies as merely unjust support for fossil fuel interests. We concur that the existence of a compensation policy by no means ensures a just coal phase-out. To begin with, there is no consensus on what makes a transition just, who has a right to participate, or how the outcomes should be assessed. Our results provide an empirical input for such debates by showing that the main beneficiaries of compensation policies are coal-dependent countries and regions together with affected energy companies rather than workers. Future research should investigate to which extent this distribution towards larger actors and entities aligns with principles of distributional, restorative, procedural and recognitional justice (Supplementary Note 4).

We nevertheless have decided to retain the language of just transitions because we are focusing on policies which policy-makers and societal actors themselves label as "just transition" policies. The majority of the policies we analyze are explicitly called "just transition" policies or partnerships (e.g. Spain calls its policy "Just Transition Strategy"¹⁹, Canada has instituted compensation according to recommendations from the "Just

transition task force”²⁰, the EU calls their policy “Just Transition Fund”²¹, and “Just Energy Transition Partnerships” have been launched with South Africa, Vietnam and Indonesia^{22–24}, etc.). In the few cases where the term “just transition” is not used in the title of the policies supporting coal phase-out, the term “just transition policy” is often used in the public and scientific debate to describe such policies. For example, in Germany, the phase-out support law that we consider is called the “structural development law”, however this law is referred to as Germany’s “just transition” – see for example ref²⁵.

Thus our object of study is what policy-makers and societal actors are calling just transition policies. As a result, our work sheds light on how just transitions are being negotiated in the real world and which actors are most commonly compensated which we highlight in the text now.

Additionally, in Supplementary Note 4, we discuss how the concept of just transitions has changed over time. “Just transitions” were originally promoted by labor unions to support workers either through re-training or the through the development of the local economy to restore jobs^{26,27} (which is in line with the Reviewer’s understanding). However, over time, as the concept of just transition has been adopted by different political organizations and academic communities, this notion has been expanded to consider several additional beneficiaries^{28,29}. This has given rise to an ongoing debate around what types of actors should be eligible to receive compensation under just transition schemes. For example, as the reviewer notes, some argue that companies should not be considered³⁰. Others, however, argue that fossil fuel-based companies and utilities are at risk due to low-carbon transitions and thus should be considered under just transition processes³¹.

To my mind, the comparators make the China and India estimates sound unduly high. “17 times higher than all existing compensation” sounds like a lot, but China and India account for seven times as much coal power generation as the 20 countries in Supp Table 1 (according to BP Statistical Review).

We have removed the comparison to existing compensation from the abstract and conclusion. Instead, we compare compensation for China and India to other existing financial flows such as international climate finance pledges and ODA-flows. This places our estimates of the financial burden of coal phase-out in line with Paris targets for China and India, which is likely to require international support. For example, we write in the abstract:

We find that such policies would be roughly comparable to the entire global Official Development Assistance and twice as large as climate finance pledged under the Paris accord.

We maintain the comparison of compensation for China and India to all existing compensation in our results section – we believe this comparison is informative since it compares the compensation which has already been pledged by governments to compensation which is needed to overcome the socio-political barriers in the world’s largest coal consumers.

Finally, regarding the reviewer’s concern that China’s and India’s compensations seems unduly high, our results are consistent with findings from previous studies such as refs^{32,33}

that also find that China and India are likely to bear a distributionally larger burden of coal phase-out.

The comparison with 2021 ODA implies that it is like-for-like. I suggest clarifying that ODA is an annual cost, but China/India compensation is not. The phaseout compensation is presumably over about 15 years. It is still high, but not quite as extreme as equality with ODA makes it sound.

We thank the reviewer for alerting us to the fact that our previous submission was unclear on this point. We are indeed comparing annual compensation and ODA in a “like-for-like” manner by dividing total compensation by the number of years over which phase-out is pledged (thus annualizing it). Figure 4 in the main text shows compensation per year (\$bn/year) compared to other annual flows such as ODA, fossil production subsidies, COP climate finance, and the annualized climate finance request by India’s prime minister. Supplementary Figure 4 shows cumulative flows (total compensation compared to ODA and fossil fuel production accrued over the coal phase-out period), and Supplementary Table 7 summarizes the annual data from Figure 3 and the cumulative data from Supplementary Figure 4.

To clarify this in our manuscript, we have now added “annual” before each mention of compensation in the text where this is relevant.

6. Suggested additional info that would be good to include (if easily done)

Very few of the phaseout commitments are consistent with the Paris goals. By estimating 1.5°C/2°C-consistent compensation for only China and India, there is an implication that the countries with commitments have done what is needed, and a suggestion that the primary onus is on China and India (whilst those are indeed the largest coal consumers, they are not the only ones that need to do more). It would be interesting – and presumably not too difficult with the existing methodology? – to look too at what would be needed to upgrade existing commitments to Paris-consistent, and/or achieve Paris-consistent phaseouts in other major consumers that have not agree to phaseout, notably the United States.

We thank the reviewer for this suggestion. Indeed the existing phase-out pledges are insufficient to meet the Paris goals. In a previously published paper, we find that if pledged coal phase-out rates of large coal consumers like Germany, Vietnam, Indonesia (of about 20% decadal decline per TES) diffuse globally the world would be on roughly a 2.5°C trajectory³².

We have thus added compensation estimates China and India in line with 2.5°C-IPCC scenarios to the results section (p. 13) of our manuscript and Supplementary Figures 3, 4 and 5. As mentioned in the response to a previous comment (p. 14-15 of this document), our findings that China and India may bear a distributionally larger burden of coal phase-out is in line with findings from refs^{32,33}, which we now cite in our paper.

We also appreciate the suggestions for looking at what’s necessary to upgrade existing coal phase-out pledges in major consumers and the US. This is a very interesting research question and indeed our methodology could contribute to answering this question. At the

same time, doing this analysis would require a substantial amount of new analysis and we think it is beyond the framing of our current paper.

Some smaller points:

Table 1: I suggest adding a column stating the phaseout year – this is important because of the earlier mention of 1.5°C and 2°C in the Introduction, and again the risk of implying that these phaseouts are Paris-aligned

We thank the reviewer for these suggestions. We have added the phase-out year to Table 1. Regarding the relationship of current coal phase-out pledges to Paris-consistent goals see response above to the earlier comment on page 15.

Page 7: What is the mean compensation per tCO₂? This would be a useful finding.

We thank the reviewer for this important and helpful suggestion. The average compensation per ton CO₂ is 37.5/tCO₂ (central; uncertainty range \$29-\$46/ton CO₂). We calculate this as the total of compensation divided by the sum of avoided emissions in all countries with compensation policies. We have added this to the main text, and in Figure 3. We also find that this is comparable to the findings from our regression analysis where the coefficient of compensation per tCO₂ ranges from \$27-45/t avoided CO₂ emissions across our ten best performing models. As explained above, we now triangulate our regression analysis and compensation estimates for China and India using the average compensation per tCO₂.

Page 7 and figure 2: For national versus international shares of compensation, it would be interesting to separate donor from ODA-recipient countries (or annex 1 from annex 2, or EU+OECD from ROW etc).

We appreciate the suggestion. We added a new Supplementary Table 4 where we show shares of international and national compensation for three types of countries: first, countries that are both ODA and EU donors. Second, countries that are part of the EU and as such do not receive ODA, but that are net-recipients of EU-funding (i.e., they receive more from the EU budget than they pay into it). And third, ODA recipient countries. All countries in the third group are JETP recipients. We specify which countries are EU recipients, ODA recipients, or EU and ODA donors in Supplementary Table 5.

Overall, we find that JETP-recipients receive a much higher share of international finance (87% central estimate) than all other countries combined (13% central estimate). We also added this finding to the main text. In Supplementary Note 1, we mention the caveat that JETP-funding mainly consists of loans rather than grants, which JETP-recipients have argued are not as valuable^{34,35}.

Since the regression is central to estimating the China and India compensation requirements (if I understand the methodology correctly), I suggest making the regression results clearer in the main text, rather than just in Methods (lines 637-641).

We agree with the reviewer that the regression needs to be better explained in the main text. We have now added several paragraphs to our results section (page 10 in our main text), to better describe the multivariable regression:

We estimate the relationship between avoided emissions and compensation using a multiple variable regression analysis with the amount of compensation as the dependent variable, the avoided emissions as the main independent variable and control variables reflecting the size and regional concentration of the coal sector, the national economic and state capacity, which affects the phase-out, as well as access to international funding, which can overcome limits from domestic financing (Methods, Supplementary Table 5, Supplementary Note 3).

In response to this and other reviewer comments (see pages 3-5, 8, 10-11), we have now also modified how we use the results from the regression analysis. We now triangulate our results of compensation estimates for China and India using both the average of compensation per ton of CO₂ which we observe in our empirical cases and a calculation based on our regression analysis where we control for additional variables (using our best-performing models, and their confidence interval).

7. Clarity issues

Line 13: The finding of proportionality to avoided emissions should probably add “when controlling for XXX”, as otherwise it looks like a conflict with Figure 2.

We thank the reviewer for this suggestion, and agree that Figure 3 (Figure 2 in our former manuscript) shows that avoided emissions is not strictly proportional to amount of compensation. We have adjusted the text to indicate that avoided emissions is proportional to ambition when controlling for characteristics of the coal sector and national context.

Lines 80-82: It would be clearer to say “We define this as the difference in cumulative emissions between two scenarios:...”

Thank you for this suggestion. We have adjusted the description of the avoided emissions calculation and now write:

We operationalize ambition as “avoided emissions” which we calculate as the difference in cumulative emissions between a reference scenario where coal power plants are retired as they reach the average national retirement age on the one hand and on the other hand a coal phase-out scenario in line with national pledges (Methods, ref).

Table 1: What does “Support closure of coal plants” mean?

Support for closure of coal plants means compensation to companies for example to coal power plant owners for foregone revenue, or financial support to coal companies to invest in alternatives to coal. We added Supplementary Table 2 where we describe what we mean by the labels of support type in Table 1 and new Figure 2.

Table 1: Is there a difference between “assisting laid-off workers” and “support for laid-off workers”?

Thanks for spotting this. There's not a difference. We have revised the coding of beneficiaries of compensation flows – this type of compensation is now called “Unemployment benefits and retraining”.

Table 3: For the four individual countries, the range of phaseout year refers to an uncertainty range, whereas for “all other countries” it reflects a range among actual data points. I find it confusing to give a single year of 2036, which is presumably a mean or a median; better just to give the range, or express it differently, to show that it means something different from the other four rows. It would also be nice to separate “all other countries” between Global North and Global South.

We thank the reviewer for these suggestions. We have changed the phase-out years to now show the range from earliest to latest phase-out year for “all other countries” (now in rows marked “JETP recipients” and “non-JETP recipients”).

We have also changed Table 3 to differentiate between JETP and non-JETP recipients. JETP recipients are essentially the Global South countries in our sample with coal phase-out pledges and compensation, and Global North countries are all other compensation countries. (The only other Global South country with coal phase-out and compensation is Chile, for which we could not quantify compensation and which is thus not included in this table).

Line 231: “would require using most...” implies that the amount of climate finance is fixed. I suggest changing to “would require an amount equal to...”

Thank you for this suggestion. We agree with the Reviewer. We have updated the text to refer to Paris climate finance pledges rather than international climate financing in general.

Lines 179-180 (also 552-553): The statement that avoided emissions are proportional to stranded capacity is not clear to me. Surely utilisation rates vary between countries? That said, it doesn't need to be proportional for the argument being made here: it is enough to say that retired capacity and avoided emissions are both appropriate measures of phaseout ambition. In those same lines, I would suggest avoiding the word “stranded”, which is generally taken to refer to investments that make a less-than-expected or less-than-commercial return. Better would be “retired” or “retired early”.

Many thanks for this important question. We have removed that statement from our text.

Lines 554 to 563 (and possibly also 80-81): It would be good to note that the reference scenario assumes no new coal plants are built (if I understand correctly). This is a fair assumption, although in reality coal plants are still being added in some countries, as the paper notes (lines 193-194).

Thank you for this suggestion. We now clarify that for both scenarios, we assume that the only new coal power plants which are built were “under construction” at the time of the coal phase-out pledge.

Lines 557 to 559: I am confused by this sentence. Since this paragraph is about the reference scenario, where retirements follow their natural course, why are pledges having an effect?

Thank you for this question. We apologize if our original formulation was not clear. This paragraph does not refer to the reference scenario, it describes how we calculate the difference between the reference and coal phase-out scenarios. We have clarified this by adding an introductory sentence to the paragraph:

We calculate the avoided emissions from coal phase-out as the difference in emissions between the reference and the coal phase-out scenarios for each country. We do this by multiplying the capacity of prematurely retired plans by the number of years between the retirement under the reference and coal phase-out scenarios and accounting for the historical national load factor as well as technology-specific efficiencies and emission rates for the thermal content of different coal types to convert avoided generation into avoided emissions (see ref. for more details).

Supplementary Table 2: The heading of column 5 appears to have been truncated. I first thought it might be number of operating coal plants but the numbers are too high for that (e.g. Germany has 60, Vietnam 25, according to Global Energy Monitor); is it individual units of power plants? As of what date?

Thank you for alerting us, the heading indeed became truncated- the number refers to power plant units for the year in which each country made the phase-out commitment, from the S&P database³⁶. We have now switched the numbers to first report GW, and then the number of units in brackets, since the amount of installed capacity is more important for our analysis.

Reviewer #2 (Remarks to the Author):

The article provides an updated analysis of compensation levels directed at regions and actors affected by coal phase-out policies in countries with coal phase-out targets. A database is developed to perform an analysis of the effect of compensation plans for the realization of ambitious coal-phase out targets that are related to the Paris Agreement's objectives (1.5-2°C). The authors conclude that compensation is crucial for the feasibility of ambitious coal phase-out policies. Hence, the underlying puzzle that the article attempts to address seems to be the tensions between tempo and costs in the energy transition, i.e. how much will it cost to speed up the transition away from coal.

We would like to thank the reviewer for the summary and their time in giving us feedback on the manuscript which have been very helpful in improving our work.

My main concern is that the article confines itself to considering compensation funding as the only just transition policy measure worth investigating. Given the article's reliance on quantitative methods, it is to a certain degree understandable that the authors need numbers to measure the effects that they are interested in. However, previous studies that discuss the tensions between various concerns in energy transitions – such as costs, tempo, and justice/fairness – point to several additional factors beside compensation as crucial for the political feasibility of fossil fuel phase-out, including distributional justice, procedural justice and restorative justice (McCauley & Heffron, 2018; Trencher et al. 2020; Gürtler et al., 2021; Healy & Barry, 2017; Isoaho & Markard, 2020; Leipprand & Flachsland, 2018; Rentier et al., 2019; Harrahill & Douglas, 2019; Green and Gambhir 2020). While not suggesting to upend the current article completely, I would like to see more reference to these important contributions to the just transition literature when the authors discuss why they have chosen to confine their analysis to compensation measures only.

We thank the reviewer for this valuable suggestion regarding the connection of our research to existing debates in the just transition literature including very valuable references. We have made several revisions in response to this comment.

First, we have re-worked the Main text (primarily 'Discussion and conclusion' section) to highlight the connection between our research and the broader literature on Just Transitions and to explain why we focus on compensation policies. In particular, we relate our analysis to wider debates about distributional issues, i.e. to what extent, and how, costs and benefits of coal phase-out are distributed across different groups of affected actors:

Compensation policies can be justified not only on the basis of their political expediency making transitions feasible, but also on ethical grounds of ensuring 'just transitions'. However, an opposite position also exists which views compensation policies as merely unjust support for fossil fuel interests. We concur that the existence of a compensation policy by no means ensures a just coal phase-out. To begin with, there is no consensus on what makes a transition just, who has a right to participate, or how the outcomes should be assessed. Our results provide an empirical input for such debates by showing that the main beneficiaries of compensation policies are coal-dependent countries and regions together with affected energy companies rather than workers. Future research should investigate to which extent this distribution towards larger actors and entities

aligns with principles of distributional, restorative, procedural and recognitional justice (Supplementary Note 4). Two particularly pressing policy questions as just transition policies shift from formulation to implementation is how to ensure good governance of such policies and to what extent compensation covers the “real” costs for different actors, including under the case of international compensation paid as grants versus as loans (Supplementary Note 1).

We also added a Supplementary Note on conceptions of justice and how these relate to our results (Supplementary Note 4). We also extended our analysis on the beneficiaries of compensation related to different types of support (Figure 2). As we now discuss in the text, we believe that our analysis can provide an empirical basis for future research to examine the adequacy of compensation policies to address distributional, recognitional and restorative justice concerns.

The article should relate compensation measures to the policy priorities arising from distributional justice, procedural justice and restorative justice concerns in just transition policy processes, for instance explaining why re-skilling or public investments in R&D are important compensation-relevant policy measures in affected regions in some countries.

Many thanks for this suggestion. We now expand on policy priorities arising from different justice concerns in our new Supplementary Note 4 on just transitions. We write, for example:

The distribution of costs and benefits of coal phase-out can relate to certain countries or regions bearing the brunt of losses in jobs and tax revenues, to companies needing to close, or to consumers facing higher electricity cost. The types of costs may also differ across contexts- for some countries in the Global South, coal phase-out is likely to relate to the challenge of providing universal access to clean energy and electricity, which is not a major concern in the Global North.

As mentioned in our response to the previous comment, in our main text, we discuss how our analysis can further support future work on how compensation policies for coal phase-out is informed by concepts of distributional, procedural and recognitional justice in the conclusion.

Another concern, related to the above, is that the results provided from the article’s analysis is quite unsurprising. An overview of compensation levels cross-checked with coal phase-out targets can give us a cursory insight into how much money policymakers in each country are willing to spend, but unfortunately produces no insights into why and how coal phase-out policies has come about in individual countries. Specifically, the analysis is not able to answer the pertinent concern that the authors themselves raise, namely that the socio-political feasibility of rapid coal phase-out is unclear.

We thank the reviewer for this important question. While it may not be surprising that governments are willing to spend money to compensate actors affected by coal phase-out, that compensation policies exist in all countries with policy-driven coal phase-out and more ambitious coal phase-out plans and that there is relative consistency between the carbon

price of compensation is surprising and has significant policy implications. As we highlight in the discussion:

Our results have direct implications for both domestic policies and international agreements regarding just energy transitions. Among diverse policies to facilitate coal phase-out, our research suggests that compensating affected actors is essential, especially in case of large-scale and rapid coal phase-out. We also find that as a rule, the more ambitious the coal phase-out pledge the larger is the compensation amount, yet the size of compensation per ton of avoided emissions is comparable to the carbon price within the EU ETS. The latter is interesting, because it suggests that societies can either pay to emit CO₂, or cut emissions and compensate affected actors with approximately the same price tag, the latter option being more attractive for the climate. This casts compensation as not only necessary but also a rationally expected policy.

Our work thus advances scientific understanding of the socio-political feasibility of coal phase-out by putting a dollar amount on the costs of making coal phase-out not only techno-economically feasible but also socio-politically feasible. We also describe this in the main text:

By mapping coal phase-out compensation in the real-world, we provide the first estimate of the cost of achieving socio-political feasibility of coal phase-out.

The Reviewer also raises interesting questions, about how these processes unfold and why some countries pursue coal phase-out and others don't (rather than about the outcome of such processes which is the object of our study).

The first question (how phase-out policies are negotiated) is the subject of an emerging body of literature which maps the processes behind coal phase-out processes – who is involved and who is not – and how the negotiations proceed^{25,37,38}. One really interesting question related to this literature is whether and how compensation is affected by the negotiation process. We hope that as this literature grows, our work can be used to test the effect of these negotiation processes on the eventual architecture and level of compensation. We now highlight this possibility in the conclusion (and also Supplementary Note 4):

Our results provide an empirical input for such debates by showing that the main beneficiaries of compensation policies are coal-dependent countries and regions together with affected energy companies rather than workers. Future research should investigate to which extent this distribution towards larger actors and entities aligns with principles of distributional, restorative, procedural and recognitional justice (Supplementary Note 4).

For the second question (why coal phase-out is pursued in some countries?), there is a literature identifying what enables and blocks national coal phase-out^{10,32,39–42}. Our work builds on this literature since the same factors which influence coal phase-out may influence compensation associated with coal phase-out, which we now also describe in the main text:

In addition to avoided emissions, the size of compensation is likely to be influenced by other factors which affect coal phase-out such as vested interests and institutional capacity.

By quantifying the policy effort needed to phase-out coal in difficult contexts, our analysis contributes to a better understanding of the policy effort required to overcome barriers to coal phase-out.

More attention to the key dimensions of just transition policies (distributional, procedural, restorative) as mentioned above will allow the authors to explore their findings in a more nuanced way.

We thank the reviewer for their valuable suggestions regarding the incorporation of different dimensions of justice.

In our revised manuscript, we reflect on how our results relate to different justice dimensions in the conclusion and in a new Supplementary Note 4 on just transitions, as also described in our response to an earlier reviewer comment (pages 20-21). Additionally, in response to this and another comment from Reviewer 1 (p. 2), we also extended our analysis of the beneficiaries of compensation.

Moreover, meaningful comparison across countries is also restricted with the current analytical set-up because no attention is given to varieties of state capacities (Meckling and Nahm 2021) or varieties of capitalism (Hall and Soskice 2001) beyond the quite shallow democracy/non-democracy distinction included. My recommendation is that the authors include reference to such crucial differences in the policymaking processes across the countries included in the analysis, perhaps finding a way to measure differences (or discuss more qualitatively/empirically) in a way that could add substantially to the insights into why countries have different approaches to coal-phase out policies.

We thank the reviewer for this valuable comment. We understand the Reviewer's concern about different aspects of the policymaking process. In this revision, we have replaced the general measure of polyarchy with variables which measure different aspects of state capacity and which have been shown in the literature on coal phase-out to be good predictors of coal phase-out^{9,10}:

- Hanson and Sigman's (HS) index incorporates several dimensions of capacity, and has been shown to be a robust predictor of coal phase-out⁷.
- The Government Effectiveness Index from the World Bank as part of its Worldwide Governance Indicators and has been shown to be a good predictor of coal phase-out pledges.³⁹

In terms of the policymaking process and more generally the national context of coal phase-out, we have also added a variable which measures a country's access to international funding since international support can help overcome limitations in national state capacity and also provide pressure on national governments to phase-out coal. This is supported by our observation that compensation as a proportion of GDP generally appears to exhibit a ceiling which can be overcome with international support.

We also considered the state capacities indicator from Meckling and Nahm 2021 suggested by the reviewer, but the authors focus on a few “advanced industrialised countries”⁴³ which means this indicator was not available for most countries in our dataset and thus not suitable for use in our regression analysis which covers both advanced and emerging economies.

With regards to the varieties of capitalism concept, we also attempted to incorporate this into our analysis. Unfortunately, the application of the VoC concept beyond OECD countries has been limited, which limits its suitability for comparison of countries across wide economic and socio-political divides. We did identify a study which applies the VoC framework to 61 countries⁴⁴ – including 29 out of the 43 countries with coal phase-out pledges in our study.

While this means that we cannot include VoC in our multiple variable regression analysis, we conduct a qualitative analysis where we compare countries with coal phase-out pledges and with or without compensation policies with VoC classifications from ref⁴⁴ to see whether certain types of VoC are more, or less, likely to have coal phase-out pledges or compensation policies. We find that countries with coal phase-out pledges fall under six of the nine groups.

Based on this analysis, we conclude that the classification of countries by VoC does not seem to significantly affect compensation policies for coal phase-out in our current sample. While surely the VoC influences how compensation policies are negotiated, we don’t find a clear signal in the amount of compensation pledged, or in whether countries have coal phase-out pledges or compensation policies in the first place. For example, among the six countries that are classified as liberal market economies, two do not have a coal phase-out pledge at all (Australia and the US), two have a coal phase-out pledge but no compensation (the UK and New Zealand), and two have coal phase-out pledges and compensation (Canada and Ireland). Further research is thus required to study the relationship between the effects of government typologies on compensation for coal phase-out negotiations and outcomes in more depth.

I hope these comments can be useful for the authors in the review process!

We would like to thank the reviewer again for their insightful comments and helpful suggestions. We hope our revisions have addressed the Reviewer’s concerns.

Reviewer #3 (Remarks to the Author):

This study offers an important and original contribution to the rapidly increasing literature on coal phase-outs. The study points to a set of clear conclusions that appear well supported by the data. The conclusions also have important implications. I believe that the following comments could help improve the manuscript.

We would like to thank the reviewer for taking the time to read our manuscript and provide helpful suggestions.

Title: You might consider adding a hint about what you found in the this paper. You might also considering answering: Is the socio-technical cost affordable, feasible etc? Is more effort required?

Thank you for this suggestion- we agree that the title could be adjusted to better reflect the content of the paper. We have revised the title to be:

Compensating affected parties necessary for rapid coal phase-out but expensive if extended to major emitters

Introduction

The authors do a good job at summarizing the situation studied.

We would like to thank the reviewer for their positive feedback of our work.

Financial compensation packages: The authors do not spend many words defining that such packages might entail. Since they would take many shapes and forms, a few more words should be devoted to unpacking this generic approach so that novices might understand.

We thank the Reviewer for this feedback and agree that the notion of compensation policies needs to be well-defined. We added a definition on page 2 of the main text:

We define [compensation policies] as publicly-financed financial transfers to support actors negatively affected by coal phase-out including workers of coal power plants and mines; companies which own and operate coal power infrastructure; and countries and regions dependent on coal (Supplementary Table 2).

A clearer statement about the research gap should be given.

We agree with the reviewer that a clearer statement of the research gap is needed. We have added several sentences to define the research gap in the introduction:

For example, Germany famously pledged over €40 billion to coal dependent regions, companies and workers as part of its coal phase-out negotiations and a coalition of Global North countries signed Just Energy Transition Partnerships (JETPs) with several emerging economies to support their coal phase-out efforts (Supplementary Note 1). Despite their growing prevalence, there has been little quantitative or comparative analysis of such policies. Is compensation necessary for coal phase-out? How much does compensation cost? And what type of support do compensation policies offer?

This relates to a broader policy question of the cost of extending compensation approaches to emerging and developing economies, where the bulk of the coal fleet is located:

Our analysis also sheds light on the potential cost of extending compensation to major coal consumers to accelerate coal phase-out. The COP26 Accord calls for “targeted support [for coal phasedown] to the poorest and most vulnerable in line with national circumstances and recognizing the need for support towards a just transition”. What financial flows would be required to expand international compensation to countries with the biggest coal fleets for a global coal phase-out consistent with the Paris targets?

The first sentence of the methods (Building a database...) could be moved to the introduction and expanded.

Many thanks for this comment. We have moved this text to the Introduction and edited it slightly:

Despite [the] growing prevalence [of compensation policies], there has been little quantitative or comparative analysis of such policies. Is compensation necessary for coal phase-out? How much does compensation cost? And what type of support do compensation policies offer?

See the type on page 2 (“I”)

Thanks for alerting us to this. We have corrected it.

Page 2: The objective of the method “To evaluate the intensity and speed of national coal phase-out commitments” should be made clearer, as I had to read well into the paper before I saw how this connected to your main theme.

Many thanks for this comment. We have reformulated how we describe the method and its relationship to our overall aim in the main text:

We also investigate whether a more ambitious coal phase-out is associated with higher compensation. The ambition of a coal phase-out pledge reflects how large and young the prematurely retired coal phase-out capacity is as well as how fast the phase-out is planned to take place. We consider a coal phase-out pledge as more ambitious when the phased out coal power capacity is larger and/or younger and/or scheduled to be shut down faster. We operationalize ambition as “avoided emissions” which we calculate as the difference in cumulative emissions between a reference scenario where coal power plants are retired as they reach the average national retirement age on the one hand and on the other hand a coal phase-out scenario in line with national pledges (Methods, refs.).

Findings: These are clear and easy to follow.

We thank the reviewer for their positive feedback.

Page 7-8: We test the possibility of predicting compensation .. I was unsure of the meaning of this statement. I had to check the supplementary material to see that you conduct a regression

to to predict the size of the compensation and not just the presence of compensation packages. This should be made clearer in the main text.

Thank you for this important suggestion. In response to this, as well as earlier reviewer comments (see page 16-17), we have revised the main text to include a description of the multiple variable regression analysis:

We estimate the relationship between avoided emissions and compensation using a multiple variable regression analysis with the amount of compensation as the dependent variable, the avoided emissions as the main independent variable and control variables reflecting the size and regional concentration of the coal sector, the national economic and state capacity, which affects the phase-out, as well as access to international funding, which can overcome limits from domestic financing (Methods, Supplementary Table 5, Supplementary Note 3).

Gt of avoided emissions in tables. It should be made clear if this is carbon or CO₂ (or CO₂e).

Many thanks for catching this. We have specified that avoided emissions relate to Gt (or Mt) of CO₂ in all tables.

Table 3. I was unsure to how interpret this analysis because you start this section with a focus on India and China but include Indonesia and Vietnam. The logic for including these other countries in Table 3 should be explained in the note.

We thank the reviewer for this question, and agree that this should be clearer.

We have revised Table 3 and now divide countries in two groups: JETP countries and non-JETP countries. This division allows us to highlight international finance to major coal consumers, which are most relevant to China and India.

We have also added a sentence on this in the main text:

The two JETP-recipients that can be considered the closest analogues to China and India are Indonesia and Vietnam: both are major coal consumers and have emerging economies.

Conclusion: After reading your paper, when I arrived at the conclusion I was unsure of the evidence that supports the argument “Our research suggests that compensating affected actors is necessary for accelerated coal phase-out” and “By comparing coal phase-out in countries which have compensation plans with those that don’t, we show that compensation policies are essential to realizing premature retirement of coal (abstract). I had to re-read the paper to find the evidence and found it as follows: “We find that all countries with ambitious coal phase-out commitments also plan compensation for coal phase-out (Figure 1). The idea here is that countries with compensation plans are able to realise the largest avoided emissions relative to countries that have a phase-out plan but don’t have compensation plans. This is an extremely important finding and a few more words should be devoted to explaining the figure and making sure that the reader does not miss this finding and the evidence/logic underpinning this.

Thank you for this valuable feedback. We have significantly restructured the text to better highlight this point. In our previous manuscript, we briefly discussed this finding in the section “Estimating and mapping financial flows for compensation of coal phase-out”. However, we agree with the reviewer that this essentially buried one of our main findings. We have thus added a new section to our results entitled “All countries with politically-accelerated coal phase-out and large coal fleets have compensation policies”. In this section, we now state:

We also investigate whether a more ambitious coal phase-out is associated with higher compensation. The ambition of a coal phase-out pledge reflects how large and young the prematurely retired coal phase-out capacity is as well as how fast the phase-out is planned to take place. We consider a coal phase-out pledge as more ambitious when the phased out coal power capacity is larger and/or younger and/or scheduled to be shut down faster. We operationalize ambition as “avoided emissions” which we calculate as the difference in cumulative emissions between a reference scenario where coal power plants are retired as they reach the average national retirement age on the one hand and on the other hand a coal phase-out scenario in line with national pledges (Methods, refs.).

We find that all countries with large coal fleets (≥ 20 GW installed capacity) and sufficiently ambitious coal phase-out pledges (≥ 200 Mt avoided CO_2) have compensation policies (Figure 1, Supplementary Table 3). The five countries (South Korea, Poland, Indonesia, Vietnam, and Germany) with the most ambitious coal phase-out pledges and largest coal fleets each plan compensation $> \$10$ billion and account for over 95% of today’s compensation. Most of these countries also have coal mining. Countries with smaller coal fleets (≤ 15 GWe) and no or little coal mining have compensation policies $< \$2$ billion (18 cases) and in 20 cases no compensation (Figure 1, Supplementary Table 3).

References

1. Council of Ministers of the Republic of Bulgaria. National Recovery and Resilience Plan of the Republic of Bulgaria. Preprint at <https://nextgeneration.bg/upload/71/BG+RRP+EN.pdf> (2022).
2. Johnston, R., Jones, K. & Manley, D. Confounding and collinearity in regression analysis: a cautionary tale and an alternative procedure, illustrated by studies of British voting behaviour. *Qual. Quant.* **52**, 1957–1976 (2018).
3. Cherp, A., Vinichenko, V., Tosun, J., Gordon, J. A. & Jewell, J. National growth dynamics of wind and solar power compared to the growth required for global climate targets. *Nat Energy* **6**, 742–754 (2021).
4. Eurostat. Glossary:Shannon evenness index (SEI) - Statistics Explained. [https://ec.europa.eu/eurostat/statistics-explained/index.php?title=Glossary:Shannon_evenness_index_\(SEI\)#:~:text=The%20Shannon%20evenness%20index%2C%20abbreviated,maximum%20\(h%20\(m\)\)](https://ec.europa.eu/eurostat/statistics-explained/index.php?title=Glossary:Shannon_evenness_index_(SEI)#:~:text=The%20Shannon%20evenness%20index%2C%20abbreviated,maximum%20(h%20(m)).). (2018).
5. Pai, S., Emmerling, J., Drouet, L., Zerriffi, H. & Jewell, J. Meeting well-below 2°C target would increase energy sector jobs globally. *One Earth* **4**, 1026–1036 (2021).
6. Mesquita, B. B. de & Smith, A. *The dictator's handbook: Why bad behavior is almost always good politics*. (PublicAffairs, 2012).
7. Hanson, J. K. & Sigman, R. Leviathan's Latent Dimensions: Measuring State Capacity for Comparative Political Research. *J Politics* **83**, 1495–1510 (2021).
8. World Bank. Worldwide governance indicators. (2023).
9. Vinichenko, V., Cherp, A. & Jewell, J. Historical precedents and feasibility of rapid coal and gas decline required for the 1.5°C target. *One Earth* **4**, 1477–1490 (2021).
10. Brutschin, E., Schenuit, F., Ruijven, B. van & Riahi, K. Exploring Enablers for an Ambitious Coal Phaseout. *Politics Gov* **10**, 200–212 (2022).
11. iForest (International Forum for Environment, Sustainability and Technology). National report release-Korba: Planning a Just Transition for India's Biggest Coal and Power District - iFOREST - International Forum for Environment, Sustainability & Technology. <https://iforest.global/events/national-report-release-korba-planning-a-just-transition-for-indias-biggest-coal-and-power-district/> (2022).
12. He, G. *et al.* Enabling a Rapid and Just Transition away from Coal in China. *One Earth* **3**, 187–194 (2020).
13. Woody, C. & AFP. China Industrial Shift Steel Coal Overcapacity. *Business Insider* <https://www.businessinsider.com/afp-steeling-for-a-struggle-china-workers-face-turmoil-2016-4?r=US&IR=T> (2016).

14. Hao, T. There's another way to solve China's industrial overcapacity | China Dialogue. *China Dialogue* <https://chinadialogue.net/en/pollution/9510-there-s-another-way-to-solve-china-s-industrial-overcapacity/> (2016).
15. Ministry of Economic Affairs and Employment. Minister Tiilikainen: Finland to ban coal in 2029 – incentives package for faster phase-out - Ministry of Economic Affairs and Employment. <https://tem.fi/en/-/ministeri-tiilikainen-kivihiihlen-kielto-2029-kannustepaketti-nopeille-luopujille> (2018).
16. Government of Canada. Powering past coal alliance: phasing out coal. <https://www.canada.ca/en/services/environment/weather/climatechange/canada-international-action/coal-phase-out.html> (2023).
17. Czyżak, P. *et al.* Poland's planned coal monopoly – who pays the price? https://instrat.pl/wp-content/uploads/2020/12/CE_Instrat_Coal-Monopoly_3.12.2020.pdf (2020).
18. Rogelj, J. *et al.* Mitigation Pathways Compatible with 1.5°C in the Context of Sustainable Development. *Special Report on Global warming of 1.5°C (SR15)* (2018).
19. Ministry of Energy. *Estrategia de transición justa en el sector energía*. https://energia.gob.cl/sites/default/files/documentos/estrategia_transicion_justa_2021.pdf (2021).
20. Task Force on Just Transition for Canadian Coal Power Workers and Communities. *A just and fair transition for Canadian coal power workers and communities*. https://publications.gc.ca/collections/collection_2019/eccc/En4-361-2019-eng.pdf (2018).
21. European Commission. *European Semester 2020 Overview of Investment Guidance on the Just Transition Fund 2021-2027 per Member State (Annex D)*. https://ec.europa.eu/info/sites/default/files/annex_d_crs_2020_en.pdf (2020).
22. European Commission. South Africa Just Energy Transition Investment Plan. https://ec.europa.eu/commission/presscorner/detail/en/STATEMENT_22_6664 (2022).
23. Foreign & Commonwealth and Development Office. Political declaration on establishing the Just Energy Transition Partnership with Viet Nam - GOV.UK. <https://www.gov.uk/government/publications/vietnams-just-energy-transition-partnership-political-declaration/political-declaration-on-establishing-the-just-energy-transition-partnership-with-viet-nam> (2022).
24. European Commission. Just Energy Transition Partnership with Indonesia. https://ec.europa.eu/commission/presscorner/detail/en/IP_22_6926 (2022).
25. Gürtler, K., Beer, D. L. & Herberg, J. Scaling just transitions: Legitimation strategies in coal phase-out commissions in Canada and Germany. *Political Geogr.* **88**, 102406 (2021).
26. Young, J. Just Transition: A New Approach to Jobs v. Environment. *Work* **2**, 42–48 (1998).

27. Abraham, J. Just Transitions for the Miners: Labor Environmentalism in the Ruhr and Appalachian Coalfields. *New Political Sci* **39**, 218–240 (2017).
28. McCauley, D. & Heffron, R. Just transition: Integrating climate, energy and environmental justice. *Energ Policy* **119**, 1–7 (2018).
29. Stevis, D. & Felli, R. Planetary just transition? How inclusive and how just? *Earth Syst Gov* **6**, 100065 (2020).
30. Oei, P.-Y. Germany's coal exit law: Too late and too expansive. *BRE Review* (2020).
31. Green, F. & Gambhir, A. Transitional assistance policies for just, equitable and smooth low-carbon transitions: who, what and how? *Clim. Polic.* **20**, 902–921 (2020).
32. Vinichenko, V., Vetier, M., Jewell, J., Nacke, L. & Cherp, A. Phasing out coal for 2 °C target requires worldwide replication of most ambitious national plans despite security and fairness concerns. *Environ Res Lett* **18**, 014031 (2023).
33. Muttitt, G., Price, J., Pye, S. & Welsby, D. Socio-political feasibility of coal power phase-out and its role in mitigation pathways. *Nat Clim Change* **13**, 140–147 (2023).
34. Fitch Ratings. Debt-Funded Energy Transition Schemes Unlikely to Weigh on EM Credit Profiles. <https://www.fitchratings.com/research/sovereigns/debt-funded-energy-transition-schemes-unlikely-to-weigh-on-em-credit-profiles-11-11-2022> (2022).
35. Naudé, L. Just Energy Transition Partnership offers should come as grants, not loans | WWF South Africa. WWF <https://www.wwf.org.za/?41686/Just-Energy-Transition-Partnership-offers-should-come-as-grants-not-loans> (2022).
36. S&P Global. *World Electric Power Plants Database*. (2021).
37. Brauers, H., Hauenstein, C., Braunger, I., Krumm, A. & Oei, P.-Y. *Comparing coal commissions – What to learn for future fossil phase-outs?* <https://vpro0190.proserver.punkt.de/s/FDDJBJ2QCS3g8rg> (2022).
38. Cha, J. M. A just transition for whom? Politics, contestation, and social identity in the disruption of coal in the Powder River Basin. *Energy Res Soc Sci* **69**, 101657 (2020).
39. Jewell, J., Vinichenko, V., Nacke, L. & Cherp, A. Prospects for powering past coal. *Nat Clim Change* **9**, 592–597 (2019).
40. Lægreid, O. M., Cherp, A. & Jewell, J. Coal phase-out pledges follow peak coal: evidence from 60 years of growth and decline in coal power capacity worldwide. *Oxf. Open Energy* **2**, oiad009 (2023).
41. Blondeel, M., Graaf, T. V. de & Haesebrouck, T. Moving beyond coal: Exploring and explaining the Powering Past Coal Alliance. *Energy Res Soc Sci* **59**, 101304 (2020).

42. Jakob, M., Flachsland, C., Steckel, J. C. & Urpelainen, J. Actors, objectives, context: A framework of the political economy of energy and climate policy applied to India, Indonesia, and Vietnam. *Energy Res Soc Sci* **70**, 101775 (2020).
43. Meckling, J. & Nahm, J. Strategic State Capacity: How States Counter Opposition to Climate Policy. *Comp Polit Stud* **55**, 493–523 (2022).
44. Witt, M. A. *et al.* Mapping the business systems of 61 major economies: a taxonomy and implications for varieties of capitalism and business systems research. *Socio-Econ. Rev.* **16**, 5–38 (2017).

REVIEWER COMMENTS

Reviewer #1 (Remarks to the Author):

Thanks to the authors for their further work on the paper, which is stronger as a result. I am grateful that they have answered my comments so thoroughly, tested some of the other regression variables I suggested, and expanded the uncertainty analysis. I also like the way variables are now grouped by issue/type in the Methods.

Most of my comments have been addressed. I have just a few modest follow-ups, numbered in relation to my original comments and their breakdown into parts in the authors' response:

(1a) Thanks for the addition of figure 2, which I think enhances the value of the paper. I recommend adding "(to companies)" to the label "power plant and mine closure", both for clarity, and for consistency with the other labels.

(4d) I note the authors' comment that they have followed ref 18 in using C1 and C2 scenarios for 1.5°C, and C3 and C4 for 2°C. I don't see the definition in that paper, and it's not a great one to follow here, as it isn't really about scenarios. In my view, it is more common in the literature to focus respectively on C1 scenarios and C3 scenarios for those temperature goals – for example, this is the convention in the AR6. Since they say this selection does not make a significant difference to results, it seems an unnecessary cost to diverge from that convention. So my recommendation would be to focus on C1 and C3. Or if not, to make clear in the main text, not only in Methods, that the article takes a different approach from what many readers are likely to assume.

(5b) I would challenge the authors' statement that "there is no consensus on what makes a transition just". In my view, it is very well defined in the ILO's 2015 Guidelines, and generally it is agreed in the literature that its key components are social dialogue, social protection, retraining, investments in alternative job-creating sectors, and targeted regional/local diversification efforts. Of course the precise application of these elements varies between regions, and is sometimes contested (and this is what some of refs 62-66 and SI refs 105-115 speak to). That doesn't imply an absence of consensus about the meaning of "just transition".

I also don't really agree that the concept has changed, as the first paragraph of Supp Note 4 suggests. For example, environmental remediation was a key element in the first conceptions of just transition in the 1990s, when just transition was proposed in relation to sites that emitted toxic local pollution (it became applied to climate change much later). Recent scholarship has located just transition in wider justice

discourses, but not really changed its meaning, nor shifted the general identification of who should be beneficiaries: I think everyone agrees that just transition is about workers and affected local communities. I have not seen anyone suggest compensation to companies is part of a just transition (though incentives for them to behave in ways that create alternative jobs might be).

I therefore recommend deleting the sentence on lines 379-381.

With the exception of that sentence, I think that paragraph is a great addition. It very helpfully highlights that by focusing on compensation, the article speaks mainly to increasing the socio-political feasibility of transitions, though it is also adjacent to – and contributes to thinking on – what makes transitions just.

With that in mind, I also recommend deleting the phrase “just transition” in line 9 of the abstract: it would be more precise to describe it as a “comparative analysis of domestic and international policies that compensate actors affected by coal phase-out”.

(5c) Sorry if I was unclear. I do not think the estimated amounts for China and India are unduly high, rather just that the previous wording implied that. The numbers are high because the quantities of coal are high. I suggest adding a sentence to that effect after line 298: “This is because the amounts of needed coal closure in these two countries are so large”.

(5d) Apologies for misreading this.

A couple of small other points:

Lines 232-240: I’m not clear on the relevance of the comparison to the cost of new coal plants. It might be relevant to compare the part of compensation *that goes to the companies*, as it would be interesting to know if companies were being given more than the book value of their assets. But the broader set of compensation measures relates to far more than just the costs of the plant, so the comparator seems misplaced.

Lines 253-254: I suggest changing “climate targets” to “Paris Agreement temperature goals” (to make clear this isn’t referring to countries’ own targets).

Lines 290-291: I think the India JETP is now abandoned. See e.g. <https://www.climatechangenews.com/2023/09/13/why-india-is-rebuffing-a-coal-to-clean-deal-with-rich-nations/>

Reviewer #2 (Remarks to the Author):

I would like to congratulate the authors for making very good revisions, hence making this an important contribution to the literature!

Reviewer #3 (Remarks to the Author):

The authors have enthusiastically taken on board the comments from the previous round and made several changes that have considerably improved the paper. Methodologically, the authors have improved their regression by adding new variables. They have also improved the implications and utility of the paper by adding an analysis of how much it would cost India and China, the two countries with the largest coal plant fleets, to pursue a similar phase-out with compensation. I have some small

comments that I think can help to polish the manuscript. After this, I would be happy to endorse publication of the study.

Introduction: Line 21 “with cheaper electricity”. This expression should be sharpened to “with rapidly declining renewable electricity costs around the world” or something similar to make its intended meaning more explicit

Sentence: “But coal phase-out leaves...” The consequences listed here are risks rather than certainties and this should be made clear

P2 First mention of China and India. It should be made clear that they do not have coal phase-out plans and mention why there were chosen for analysis in just a few words

P2 Lines 55 . When you mention support for national governments, you should make it clear that this is the case of compensation from international mechanisms.

Line 153. If the list of compensation types mentioned here are all forms of “support”, then the colon should appear as follows. “These encompass support for: regional development to regional authorities, ...”

Findings and interpretation: As you acknowledge, a just transition should involve the direction of funds to the most affected. But the volume of funds should also be correlated with the amount of damage suffered. In your case, you find that the larger the avoided emissions, the larger the payment amount. This hints that justice may be at work in some way in current compensation packages from the perspective of compensation amounts. This possibility could be mentioned, bearing in mind that other dimensions in a just transition (e.g. how gets the money and how is it used) are outside the scope of the paper.

Line 368: Because the price of carbon in the EU-ETS is so volatile and changes each year, you should drop a ball park figure here

Line 378 Another risk with coal phase-out compensation, like REDD+, is that this can create a perverse incentive for developers or governments to build plants, as they might assume that any risks of financial loss in the future would be borne by other countries or the international community. You could state this briefly.

Another implication of your findings that you do not spell out much is that the tremendous cost of paying for countries to retire plants early points to the urgent need to halt new plants under construction, especially in countries with no phase-out plans. Compared to the cost of compensating plants once built, halting their construction appears attractive from an economic feasibility perspective when considering your findings.

You also should make it clear what a “non-policy-driven phase-out” is. I can understand this, but readers might not, and your meaning is not explicit.

Good luck in refining.

Dear Reviewers,

Thank you for your positive reception of our revised manuscript, as well as for your further comments which have helped us to further improve our work. In response to Reviewers 1 and 3, we have made several revisions and added several clarifications to our main text:

1. Throughout the text, we now refer to the 1.5°C and 2°C targets as the Paris temperature targets rather than the ‘climate targets’ as suggested by Reviewer 1.
2. We now clarify in the Introduction that our analysis focuses on China and India because these are the two countries with the largest coal fleets and no existing coal phase-out pledges as suggested by Reviewer 3.
3. We revised the findings section titled ‘Compensation is proportional to avoided emissions and comparable to recent carbon prices’ for clarity, in particular our discussion of compensation per GW for companies as suggested by Reviewer 1.
4. We revised the findings section titled ‘Compensation for coal phase-out in China and India would dwarf existing climate finance’, in particular our description of how we estimate compensation for China and India including our approach to IPCC scenarios in the main text (which we also expanded in the Methods) as suggested by Reviewer 1.
5. We revised the Discussion and conclusions section, in particular how we discuss just transitions and potential feedbacks between compensation policies and coal phase-out pledges. With respect to just transitions, we removed a statement on the contestation of just transitions from the Discussion and also added Supplementary Table 17 to address questions about the contestation of just transitions raised by Reviewer 1. Additionally, we reflect on the justice implications of our findings as well as how today’s compensation policies may affect the coal pipeline by creating expectations of future funding opportunities (or costs) for different actors as suggested by Reviewer 3.

Finally, we have added references to several recent developments since the last submission of our manuscript. As suggested by Reviewer 1, we now refer to a recent news report that a JETP with India may be unlikely in the short term (Chandrashekar, 2023). We also cite a report on just transition costs in India (Bhushan, 2023) and acknowledge the addition of six new members to the PPCA (PPCA, 2023) (which does not change our analysis because these new members either do not have a defined year for coal phase-out, or have no installed coal capacity – Supplementary Table 3).

Overall, we believe these changes further strengthen our manuscript by more clearly communicating the findings and their implications.

We thank the Reviewers once again for their time and look forward to your further comments. Please find a point-by-point explanation for how we responded to your comments below.

Text in black indicates original reviewer comments, text in blue indicates author responses, and text in green indicates citations from the manuscript. Please note that we have chosen a different reference style here to differentiate from references used in the main text.

Reviewer 1

Thanks to the authors for their further work on the paper, which is stronger as a result. I am grateful that they have answered my comments so thoroughly, tested some of the other regression variables I suggested, and expanded the uncertainty analysis. I also like the way variables are now grouped by issue/type in the Methods. Most of my comments have been addressed. I have just a few modest follow-ups, numbered in relation to my original comments and their breakdown into parts in the authors' response:

We thank the reviewer for their positive reception of our revision and for their further comments, which we respond to below!

(1a) Thanks for the addition of figure 2, which I think enhances the value of the paper. I recommend adding "(to companies)" to the label "power plant and mine closure", both for clarity, and for consistency with the other labels.

We thank the reviewer for their suggestion, and have revised the label to now state "Power plant and mine closure (to companies)".

(4d) I note the authors' comment that they have followed ref 18 in using C1 and C2 scenarios for 1.5°C, and C3 and C4 for 2°C. I don't see the definition in that paper, and it's not a great one to follow here, as it isn't really about scenarios. In my view, it is more common in the literature to focus respectively on C1 scenarios and C3 scenarios for those temperature goals – for example, this is the convention in the AR6. Since they say this selection does not make a significant difference to results, it seems an unnecessary cost to diverge from that convention. So my recommendation would be to focus on C1 and C3. Or if not, to make clear in the main text, not only in Methods, that the article takes a different approach from what many readers are likely to assume.

We thank the reviewer for coming back to this important comment!

Let us first start with apologising for the confusion related to reference 18 and clarify that ref. 18 referred to the reference in the Reviewer responses (Riahi et al., 2022) (which is focused on scenarios) and not the reference in the main text which as the Reviewer notes is clearly not.

We appreciate and follow the reviewer's last suggestion to highlight this approach more clearly in our main text, and have added a sentence to our results section:

"First, we calculate the avoided emissions for China and India in 1.5°C-compatible (IPCC AR6 C1 and C2 categories) and 2°C-compatible pathways (C3 and C4 categories) (Methods)."

We have also added brief sentences explaining our approach to the captions of Table 3 and Figure 4, where scenario data is used.

Finally, we conducted a brief review of literature which uses IPCC scenarios to analyse the Paris Agreement temperature targets to verify our approach, and have found a wide range of interpretations. Some authors take a very narrow view and only view a subset of C1-pathways as Paris-consistent (Schleussner et al., 2022). Others argue that all pathways under C1 and C2 are consistent with 1.5°C of warming (Gambhir et al., 2023; Riahi et al., 2021) while still others consider a subset of pathways under C2 and C3-categories consistent with 1.5°C (Iyer et al., 2022). Finally, some use C1 and C2 categories as 1.5°C-consistent as well as C3 and C4 categories as 2°C-consistent (Vinichenko, Jewell, et al., 2023; Vinichenko, Vetier, et al., 2023). Thus, we position ourselves on the broader end of this range of IPCC-pathway interpretations - despite which our cost estimates for China and India remain relatively high.

(5b) I would challenge the authors' statement that "there is no consensus on what makes a transition just". In my view, it is very well defined in the ILO's 2015 Guidelines, and generally it is agreed in the literature that its key components are social dialogue, social protection, retraining, investments in alternative job-creating sectors, and targeted regional/local diversification efforts. Of course the precise application of these elements varies between regions, and is sometimes contested (and this is what some of refs 62-66 and SI refs 105-115 speak to). That doesn't imply an absence of consensus about the meaning of "just transition". I also don't really agree that the concept has changed, as the first paragraph of Supp Note 4 suggests. For example, environmental remediation was a key element in the first conceptions of just transition in the 1990s, when just transition was proposed in relation to sites that emitted toxic local pollution (it became applied to climate change much later). Recent scholarship has located just transition in wider justice discourses, but not really changed its meaning, nor shifted the general identification of who should be beneficiaries: I think everyone agrees that just transition is about workers and affected local communities. I have not seen anyone suggest compensation to companies is part of a just transition (though incentives for them to behave in ways that create alternative jobs might be). I therefore recommend deleting the sentence on lines 379-381.

Thank you for these important questions regarding the history and current debates around the just transition concept! We generally agree with the reviewer that the overall understanding is similar across organisations and disciplines while exact definitions and principles of implementation vary. We have thus removed the statement on the contested nature of the just transition from the Discussion and conclusion section.

We have however added our new Supplementary Table 17, where we summarise the main just transition definitions across a selection of four organisations, some of which include industries among beneficiaries of just transitions. An example is the European Bank for Reconstruction and Development (EBRD)'s definition: "A just transition seeks to ensure that the substantial benefits of a green economy transition are shared widely, while also supporting those who stand to lose economically – be they countries, regions, industries, communities, workers or consumers." (European Bank for Reconstruction and Development, n.d.). The recognition of companies as relevant beneficiaries to just transition policies is also echoed in the scientific literature by Green and Gambhir (2020), which recognizes companies as "adversely affected

actors” of low-carbon transitions and recommends transitional assistance policies to these actors as part of “just, equitable and politically smooth” transitions.

Regarding the question of whether the just transition concept has evolved overtime, while we agree with the reviewer that the original just transition concept was already connected to the environmental justice movement (Rector, 2018; Stevis & Felli, 2015), the application of just transition beyond the labor-environment nexus has developed and expanded over the last four decades. The consideration of broader societal issues such as energy access, gender equality, and inequalities between the Global North and Global South, which today are commonly discussed in the scientific and policy debates around just transitions (see for example Table 1 above), were not within the scope of the just transition concept as originally coined by US and Canadian labour unions (Committee for Development Policy, 2023; McCauley & Heffron, 2018; Newell & Mulvaney, 2013; Stevis & Felli, 2015). We have thus edited our Supplementary Note 4 to better reflect the current contestations and historical evolution of the just transition concept.

(5b ctd) With the exception of that sentence, I think that paragraph is a great addition. It very helpfully highlights that by focusing on compensation, the article speaks mainly to increasing the socio-political feasibility of transitions, though it is also adjacent to – and contributes to thinking on – what makes transitions just. With that in mind, I also recommend deleting the phrase “just transition” in line 9 of the abstract: it would be more precise to describe it as a “comparative analysis of domestic and international policies that compensate actors affected by coal phase-out”.

We are glad that the Reviewer thinks the new paragraph adds to the article! We also appreciate and follow the reviewer’s suggestion to remove the term “just transition” from line 9 of the abstract.

(5c) Sorry if I was unclear. I do not think the estimated amounts for China and India are unduly high, rather just that the previous wording implied that. The numbers are high because the quantities of coal are high. I suggest adding a sentence to that effect after line 298: “This is because the amounts of needed coal closure in these two countries are so large”.

We thank the reviewer for the clarification and suggestion! We have opted to not add the sentence suggested by the reviewer, since we already highlight that China and India are the two countries with the largest coal fleets globally (in the Introduction and on page 11), and describe that the required coal phase-out ambition for these countries is also much higher than in countries with existing pledges (in the second paragraph of the results). This is also shown in Table 3, where we compare avoided emissions required in different countries/country groups.

(5d) Apologies for misreading this.

We are happy we could clarify the relevant passage!

A couple of small other points:

Lines 232-240: I'm not clear on the relevance of the comparison to the cost of new coal plants. It might be relevant to compare the part of compensation *that goes to the companies*, as it would be interesting to know if companies were being given more than the book value of their assets. But the broader set of compensation measures relates to far more than just the costs of the plant, so the comparator seems misplaced.

We thank the reviewer for this important question! We have added this benchmark for completeness, as it has been used in the emerging literature on coal phase-out costs (Tiedemann & Müller-Hansen, 2023). Similarly to what the reviewer suggests, Tiedemann and Müller-Hansen (2023) compare German *coal phase-out compensation to companies only* to the capital cost of new coal-fired power plants in Europe, and finds that this compensation is lower than the cost of a new coal plant. In our study, we find that six countries have compensation to coal power and/or mining companies (Figure 2), and in all cases *compensation to companies only* would be lower than the cost of a new coal power plant.

At the same time, we recognize that capital cost is not the same as the book value of power plants since it does not capture depreciation over time and does not factor in electricity prices, capacity markets, or carbon prices. In fact, even though the compensation in Germany was less than the cost of a new power plant, the German government was criticized for overcompensating power plant owners relative to the value of their power plants (Brown, 2021). We have added a sentence in the main text to reflect this:

“In the case of Germany, while the overall compensation is greater than the cost of a new coal power plant, compensation to companies supporting power plant and mining closure is less; nevertheless, some have criticized the German government for overcompensating companies relative to the value of the retired coal plants.”

The “book value” of the plants is likely one of the reasons that avoided emissions is a better predictor for compensation for coal phase-out than power plant capacity. By factoring in power plant age, and assumptions on expected lifetimes and capacity factors, it measures prematurely-retired generation. We have thus added a sentence to the text that indicates this:

“We also calculate the average compensation per GW of installed coal capacity across all countries as \$0.8 billion/GW (uncertainty range \$0.7-1.1 billion/GW), which is generally below the cost of new coal power capacity in Europe. This is to be expected since most countries plan to retire aging power plant fleets, which have already depreciated in value. The impact of the age of coal power plant fleets is likely why avoided emissions is a better predictor for compensation than installed capacity.”

Lines 253-254: I suggest changing “climate targets” to “Paris Agreement temperature goals” (to make clear this isn't referring to countries' own targets).

We agree with the reviewer’s suggestion and now refer to the “Paris Agreement temperature targets”.

Lines 290-291: I think the India JETP is now abandoned. See e.g. <https://www.climatechangenews.com/2023/09/13/why-india-is-rebuffing-a-coal-to-clean-deal-with-rich-nations/>

We thank the reviewer for highlighting this development and have adjusted the language regarding the JETP-discussions with India, where we now also refer to a recent report by an Indian organisation which underlines the high economic burden of coal phase-out and likely requirement of international funding sources for the case of India:

“Given the high cost for China and India, international funding might be required for coal phase-out compensation (as ref. [Bhushan 2023] argues for India) though there is no such negotiation process for China and recent reports indicate that a JETP-type agreement between India and donor countries is unlikely.”

Reviewer 2

I would like to congratulate the authors for making very good revisions, hence making this an important contribution to the literature!

We thank the reviewer for their positive reception of our revisions!

Reviewer 3

The authors have enthusiastically taken on board the comments from the previous round and made several changes that have considerably improved the paper. Methodologically, the authors have improved their regression by adding new variables. They have also improved the implications and utility of the paper by adding an analysis of how much it would cost India and China, the two countries with the largest coal plant fleets, to pursue a similar phase-out with compensation. I have some small comments that I think can help to polish the manuscript. After this, I would be happy to endorse publication of the study.

We thank the reviewer for their positive reception of our previous revision, as well as for their further comments!

Introduction: Line 21 “with cheaper electricity”. This expression should be sharpened to “with rapidly declining renewable electricity costs around the world” or something similar to make its intended meaning more explicit

We thank the reviewer for this suggestion and have edited the relevant sentence:

“Phasing-out coal is one of the most urgent climate mitigation measures and recent declines in the cost of solar and wind power make it techno-economically feasible.”

Sentence: “But coal phase-out leaves...” The consequences listed here are risks rather than certainties and this should be made clear

Thank you for this comment! We agree with the reviewer that the consequences listed do not necessarily arise to the same extent in all cases of coal phase-out, and have edited the sentence:

“But coal phase-out risks stranding assets, triggering backlash from coal workers and companies, as well as causing socio-economic hardship for coal-dependent regions and electoral losses for politicians.”

P2 First mention of China and India. It should be made clear that they do not have coal phase-out plans and mention why they were chosen for analysis in just a few words

Thank you for this suggestion! We have added a sentence to clarify that China and India do not have coal phase-out pledges and are indeed the countries with the biggest coal fleets:

“Even though neither China nor India has pledged to phase-out coal, they have the largest coal fleets which make their coal policies critical to achieving the Paris temperature targets.”

P2 Lines 55 . When you mention support for national governments, you should make it clear that this is the case of compensation from international mechanisms.

We thank the reviewer for this suggestion, but have opted not to add the suggested information in the Introduction section. We find that this information is better placed in our discussion of international and domestic compensation on pages 6-7, and we also show the distribution of domestic versus international funding across the different types of support in Supplementary Figure 2.

Line 153. If the list of compensation types mentioned here are all forms of “support”, then the colon should appear as follows. “These encompass support for: regional development to regional authorities, ...”

Thank you for this suggestion, we have rephrased the sentence as follows:

“Compensation policies encompass support for five types of measures: regional development to regional authorities or SMEs; power plant and mining closure; renewables capacity and infrastructure development; and unemployment benefits and retraining.”

Findings and interpretation: As you acknowledge, a just transition should involve the direction of funds to the most affected. But the volume of funds should also be correlated with the amount of damage suffered. In your case, you find that the larger the avoided emissions, the larger the payment amount. This hints that justice may be at work in some way in current compensation packages from the perspective of compensation amounts. This possibility could be mentioned, bearing in mind that other dimensions in a just transition (e.g. how gets the money and how is it used) are outside the scope of the paper.

We appreciate the reviewer’s suggestion, and have added a sentence in our conclusion acknowledging that our finding that compensation amounts are generally proportional to avoided emissions indicates that most compensation is paid in contexts where actors are likely to experience stronger negative effects of coal phase-out:

“Our finding that the amount of compensation is proportional to avoided emissions indicates that compensation policies are not spurious, but rather seek to address negative impacts of coal phase-out, which are likely to be larger in the case of faster and wider retirement of coal power plants.”

Line 368: Because the price of carbon in the EU-ETS is so volatile and changes each year, you should drop a ball park figure here

We thank the reviewer for alerting us to this, and have opted to change our formulation to the plural (“recent carbon prices within the EU ETS”) since we also show the wide range of carbon prices in Figure 3.

Line 378 Another risk with coal phase-out compensation, like REDD+, is that this can create a perverse incentive for developers or governments to build plants, as they might assume that any risks of financial loss in the future would be borne by other countries or the international community. You could state this briefly.

Thank you for this insightful suggestion! We agree and have implemented by adding a sentence to the discussion and conclusion section, also in connection to your following suggestion.

“Today’s compensation policies may also send signals either encouraging or discouraging coal power expansion depending on whether the relevant governments and companies expect to be donors or recipients of future compensation.”

Another implication of your findings that you do not spell out much is that the tremendous cost of paying for countries to retire plants early points to the urgent need to halt new plants under construction, especially in countries with no phase-out plans. Compared to the cost of compensating plants once built, halting their construction appears attractive from an economic feasibility perspective when considering your findings.

Thank you also for this suggestion, which we think relates to your previous comment on how coal phase-out compensation interacts with coal plants in the pipelines yet to be built- see the corresponding edits we made to the text in our answer to your previous comment above.

You also should make it clear what a “non-policy-driven phase-out” is. I can understand this, but readers might not, and your meaning is not explicit.

We agree that the original formulation may be unclear to some readers. We have thus adjusted the relevant sentence to now state “even in the absence of deliberate coal phase-out policies,” clarifying what can be considered non-policy-driven coal phase-out.

Good luck in refining.

Thank you for your comments and feedback!

References

- Bhushan, C. (2023). Just Transition, Just Finance: Methodology and Costs for Just Energy Transition in India. International Forum for Environment, Sustainability and Technology (iFOREST).
- Brown, S. (2021, 16. May). Germany's flawed €4.4bn lignite compensation | Ember. EMBER. <https://ember-climate.org/insights/research/germanys-flawed-lignite-assumptions/#supporting-material>
- Chandrashekhar, V. (2023, 13. September). Why India is rebuffing a coal-to-clean deal with rich nations. Climate Change News. <https://www.climatechangenews.com/2023/09/13/why-india-is-rebuffing-a-coal-to-clean-deal-with-rich-nations/>
- Committee for Development Policy. (2023). Just transition. <https://www.un.org/development/desa/dpad/wp-content/uploads/sites/45/CDP-excerpt-2023-1.pdf>
- European Bank for Reconstruction and Development. (n.d.). What is a just transition? Retrieved January 16, 2024, from <https://www.ebrd.com/what-we-do/just-transition>
- Gambhir, A., Mittal, S., Lamboll, R. D., Grant, N., Bernie, D., Gohar, L., Hawkes, A., Köberle, A., Rogelj, J. & Lowe, J. A. (2023). Adjusting 1.5 degree C climate change mitigation pathways in light of adverse new information. Nature Communications, 14(1), 5117. <https://doi.org/10.1038/s41467-023-40673-4>
- Green, F. & Gambhir, A. (2020). Transitional assistance policies for just, equitable and smooth low-carbon transitions: who, what and how? Climate Policy, 20(8), 902–921. <https://doi.org/10.1080/14693062.2019.1657379>
- Iyer, G., Ou, Y., Edmonds, J., Fawcett, A. A., Hultman, N., McFarland, J., Fuhrman, J., Waldhoff, S. & McJeon, H. (2022). The path to 1.5 °C requires ratcheting of climate pledges. Nature Climate Change, 1–2. <https://doi.org/10.1038/s41558-022-01517-z>
- McCauley, D. & Heffron, R. (2018). Just transition: Integrating climate, energy and environmental justice. Energy Policy, 119, 1–7. <https://doi.org/10.1016/j.enpol.2018.04.014>
- Newell, P. & Mulvaney, D. (2013). The political economy of the 'just transition.' The Geographical Journal, 179(2), 132–140. <https://doi.org/10.1111/geoj.12008>
- PPCA. (2023, 3. December). COP28 opens with remarkable international actions on coal phase-out - PPCA. <https://poweringpastcoal.org/news/cop28-opens-with-remarkable-international-actions-on-coal-phase-out/>

- Rector, J. (2018). The Spirit of Black Lake: Full Employment, Civil Rights, and the Forgotten Early History of Environmental Justice. *Modern American History*, 1(1), 45–66.
<https://doi.org/10.1017/mah.2017.18>
- Riahi, K., Bertram, C., Huppmann, D., Rogelj, J., Bosetti, V., Cabardos, A.-M., Deppermann, A., Drouet, L., Frank, S., Fricko, O., Fujimori, S., Harmsen, M., Hasegawa, T., Krey, V., Luderer, G., Paroussos, L., Schaeffer, R., Weitzel, M., Zwaan, B. van der, ... Zakeri, B. (2021). Cost and attainability of meeting stringent climate targets without overshoot. *Nature Climate Change*, 1–7. <https://doi.org/10.1038/s41558-021-01215-2>
- Riahi, K., Schaeffer, R., Arango, J., Calvin, K., Guivarch, C., Hasegawa, T., Jiang, K., Kriegler, E., Matthews, R., Peters, G., Rao, A., Robertson, S., Sebbit, A. M., Steinberger, J., Tavoni, M. & Vuuren, D. van. (2022). Mitigation pathways compatible with long-term goals. In P. R. Shukla, J. Skea, R. Slade, A. A. Khourdajie, R. van Diemen, D. McCollum, M. Pathak, S. Some, P. Vyas, R. Fradera, M. Belkacemi, A. Hasija, G. Lisboa, S. Luz & J. Malley (Eds.), *Climate change 2022: Mitigation of climate change. Contribution of Working Group III to the Sixth Assessment Report of the Intergovernmental Panel on Climate Change*. IPCC.
<https://www.ipcc.ch/report/sixth-assessment-report-working-group-3/>
- Schleussner, C.-F., Ganti, G., Rogelj, J. & Gidden, M. J. (2022). An emission pathway classification reflecting the Paris Agreement climate objectives. *Communications Earth & Environment*, 3(1), 135. <https://doi.org/10.1038/s43247-022-00467-w>
- Stavis, D. & Felli, R. (2015). Global labour unions and just transition to a green economy. *International Environmental Agreements: Politics, Law and Economics*, 15(1), 29–43.
<https://doi.org/10.1007/s10784-014-9266-1>
- Tiedemann, S. & Müller-Hansen, F. (2023). Auctions to phase out coal power: Lessons learned from Germany. *Energy Policy*, 174, 113387. <https://doi.org/10.1016/j.enpol.2022.113387>
- Vinichenko, V., Jewell, J., Jacobsson, J. & Cherp, A. (2023). Historical diffusion of nuclear, wind and solar power in different national contexts: implications for climate mitigation pathways. *Environmental Research Letters*, 18. <https://doi.org/10.1088/1748-9326/acf47a>
- Vinichenko, V., Vetier, M., Jewell, J., Nacke, L. & Cherp, A. (2023). Phasing out coal for 2 °C target requires worldwide replication of most ambitious national plans despite security and fairness concerns. *Environmental Research Letters*, 18(1), 014031.
<https://doi.org/10.1088/1748-9326/acadf6>

REVIEWERS' COMMENTS

Reviewer #1 (Remarks to the Author):

Thanks to the authors for their further work on the paper, which is looking in great shape!

I am happy to recommend it for publication.